# mRNA stability fine-tunes gene expression in the developing cortex to control neurogenesis

**Lucas D. Serdar**[1], **Jacob R. Egol**[1], **Brad Lackford**[2], **Brian D. Bennett**[2], **Guang Hu**[2], **Debra L. Silver**[1,3,4] *

1 Department of Molecular Genetics and Microbiology, Duke University Medical Center, Durham, North Carolina, United States of America, 2 National Institute of Environmental Health Sciences, Durham, North Carolina, United States of America, 3 Departments of Cell Biology and Neurobiology, Duke University Medical Center, Durham, North Carolina, United States of America, 4 Duke Institute for Brain Sciences and Duke Regeneration Center, Duke University Medical Center, Durham, North Carolina, United States of America

* debra.silver@duke.edu

**Data Availability Statement:** Sequencing data has been deposited and made publicly available on GEO under accession numbers GSE281690 (RNA-seq) and GSE281693 (SLAM-seq).

## Abstract

RNA abundance is controlled by rates of synthesis and degradation. Although mis-regulation of RNA turnover is linked to neurodevelopmental disorders, how it contributes to cortical development is largely unknown. Here, we discover the landscape of RNA stability regulation in the cerebral cortex and demonstrate that intact RNA decay machinery is essential for corticogenesis in vivo. We use SLAM-seq to measure RNA half-lives transcriptome-wide across multiple stages of cortical development. Leveraging these data, we discover *cis*-acting features associated with RNA stability and probe the relationship between RNA half-life and developmental expression changes. Notably, RNAs that are up-regulated across development tend to be more stable, while down-regulated RNAs are less stable. Using compound mouse genetics, we discover CNOT3, a core component of the CCR4-NOT deadenylase complex linked to neurodevelopmental disease, is essential for cortical development. Conditional knockout of *Cnot3* in neural progenitors and their progeny in the developing mouse cortex leads to severe microcephaly due to altered cell fate and p53-dependent apoptosis. Finally, we define the molecular targets of CNOT3, revealing it controls expression of poorly expressed, non-optimal mRNAs in the cortex, including cell cycle-related transcripts. Collectively, our findings demonstrate that fine-tuned control of RNA turnover is crucial for brain development.

## Introduction

The cerebral cortex is essential for higher order functions, including cognitive reasoning, and somatosensory, motor, and visual processing. These processes rely on proper embryonic development, and defective embryonic corticogenesis can lead to neurodevelopmental disorders, including autism, schizophrenia, and intellectual disability. The developmental trajectory and underlying biological and molecular events necessary to construct the cortex during development are generally well defined [1,2]. In mice, the main neural progenitors are radial glial cells

**Funding:** This work was supported by the Extramural and Intramural Research Programs of the National Institutes of Health: F32HD107972 to L.D.S., R01NS083897, R01NS120667, R37NS110388, R01MH132089, R21NS128374 to D.L.S., and Z01ES102745 to G.H. The funders had no role in study design, data collection and analysis, decision to publish, or preparation of the manuscript.

**Competing interests:** The authors have declared that no competing interests exist.

**Abbreviations:** CC3, cleaved caspase 3; cKO, conditional knockout; CSC, codon stabilization coefficient; dcKO, double conditional knockout; GO, gene ontology; IP, intermediate progenitor; NMD, nonsense-mediated RNA decay; PCA, principal component analysis; RGC, radial glial cell.

(RGCs), which produce neurons directly, and intermediate progenitors (IPs), which are derived from RGCs and are also neurogenic (Fig 1A). The mature mammalian cortex consists of a stereotypical six-layered architecture that forms in an inside-out fashion. Neurons born early in development (~embryonic day (E)12.5 –E13.5 in mouse) occupy deep layers, while late born neurons (~E14.5 –E16.5) typically occupy superficial layers [3]. Thus, corticogenesis is a dynamic process in which progenitor potency and neuronal composition changes with development.

In line with diverse cellular processes, the transcriptomes of progenitors and neurons are dynamic during corticogenesis [4–7]. These developmental gene expression changes are controlled by transcriptional and posttranscriptional mechanisms [8,9]. RNA stability regulation is emerging as an important modulator of gene expression during cortical development. Previous work has revealed roles for core components of the RNA decay machinery, including the exosome [10] and the nonsense-mediated RNA decay (NMD) pathway [11–16]. Several studies have also highlighted the importance of additional *cis-* and *trans-*acting regulators of stability in cortical development. For example, the oscillatory RNA expression pattern of the Notch signaling pathway gene *Hes1* in cortical progenitors is maintained in part by microRNA-mediated degradation [17,18]. In addition, the abundant epitranscriptomic mark N6-methyladenosine (m$^6$A) destabilizes RNA in cortical cells and influences neurogenesis [19]. While these studies highlight fundamental roles of mRNA stability in cortical development, our understanding of this process is limited. Indeed, a survey of RNA half-lives across development is lacking, as is a mechanistic understanding of how RNA turnover impacts the developing brain.

A key regulator of mRNA stability is the CCR4-NOT deadenylase complex [20]. This multi-subunit complex catalyzes the removal of 3′ poly(A) tails from mRNAs, typically one of the first steps in mRNA degradation. CNOT3 is a non-enzymatic component of the CCR4-NOT complex and required for efficient deadenylation and mRNA degradation [21–23]. Mutations in *CNOT3* as well as another component, *CNOT1*, are linked to neurodevelopmental pathologies characterized by a spectrum of intellectual disability, speech delay, seizures, and behavioral problems [24–30]. Of note, *CNOT3* is categorized as a "High-confidence, syndromic" autism risk gene in the SFARI autism gene database. Previous work implicates CNOT3 in diverse stem cell populations, including cardiomyocytes [31,32], embryonic stem cells [33,34], and spermatogonial stem cells [35]. Despite clinical significance of *CNOT3* in neurodevelopment, its molecular function in cortical development is entirely unknown.

Here, we employ multi-pronged approaches to understand the principles and functions of RNA degradation in the developing brain (Fig 1A). We use SLAM-seq to globally measure RNA half-lives across multiple stages of mouse development. From this, we discover key features governing RNA stability in the cortex and uncover relationships between RNA turnover rates and gene expression changes across development. We then employ *Cnot3* conditional knockout mouse models to discover cellular and molecular mechanisms by which RNA decay machinery controls cortical neurogenesis. Our findings reveal the importance of RNA stability control in the developing cortex and establish the CCR4-NOT complex as a key factor contributing to proper gene expression and cellular function during corticogenesis.

## Results

### SLAM-seq defines the landscape and features of RNA stability in cortical cells

To characterize the mRNA stability landscape of the developing cortex, we performed thiol (SH)-linked alkylation for metabolic sequencing (SLAM-seq) [36] at 3 different developmental

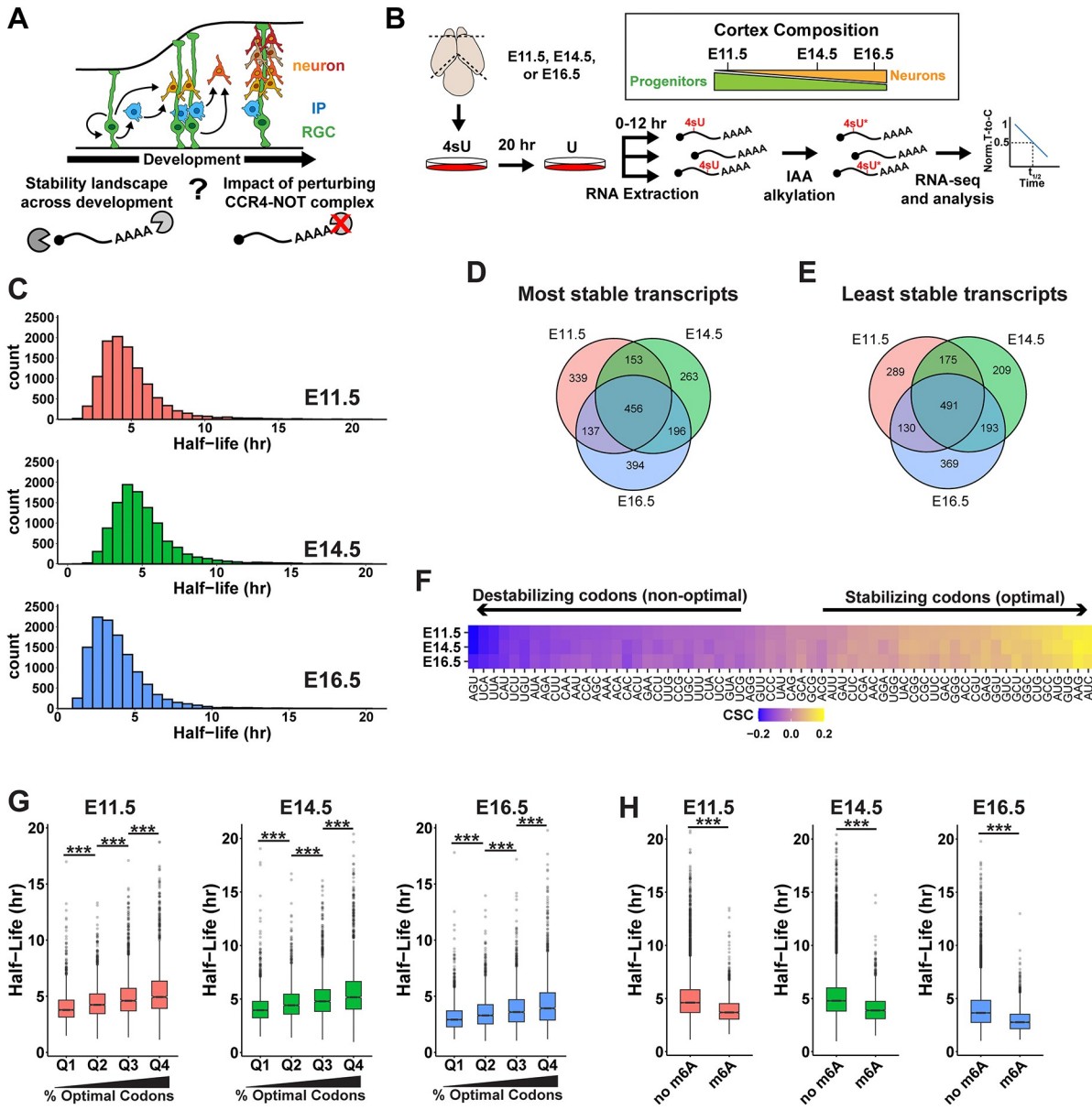

**Fig 1. The developmental landscape of cortical RNA stability.** (A) Overview of cortical development showing RGCs generating neurons directly, or indirectly through IPs, with goals of this study schematized below. (B) Approach for SLAM-seq experiment, with relative cellular composition of the cortex at the 3 developmental stages used in this study ($n$ = 3 biological replicates per stage). (C) Histograms showing distribution of RNA half-lives. (D, E) Venn diagrams for top 10% most and least stable transcripts identified at each stage. (F) CSC values for each of the 61 non-stop codons. (G) Half-lives of RNAs binned into quartiles based on percentage of optimal (positive CSC) codons. (H) Half-lives of RNAs with or without m⁶A. *** $p < 0.001$. One-way ANOVA with Tukey's HSD post hoc test (G), Wilcoxon rank-sum test (H). Underlying data for this figure can be found in S1 Data. CSC, codon stabilization coefficient; IP, intermediate progenitor; RGC, radial glial cell.

stages. We chose E11.5, E14.5, and E16.5 as representative stages of early, middle, and late neurogenesis, respectively. E11.5 cortices are predominantly composed of RGCs, while E14.5 and E16.5 cortices contain increasing numbers of IPs and neurons, along with non-RGC derived cells. Cortices from E11.5, E14.5, and E16.5 embryos were dissociated into single cell suspensions using 3 independent biological replicates per stage and cultured in vitro using conditions

permissive for proliferation [37], and 4-thiouridine (4sU) was added to culture media for 20 h to label nascent RNAs. Cells were then collected for RNA extraction 0, 2, 4, 8, and 12 h following a chase with excess unmodified uridine to monitor degradation of labeled RNAs over time (Fig 1B). Following sequencing, we obtained reproducible half-life measurements for 10,842 transcripts at E11.5, 10,671 transcripts at E14.5, and 11,832 transcripts at E16.5 (Fig 1C and S1 Table). Principal component analysis (PCA) showed that the biological replicates within each stage largely clustered together but were distinct between the stages (S1A Fig). Comparison of z-score normalized half-lives revealed high correlations between stages (r > 0.87), indicating that the relative stability of individual transcripts is highly consistent across cortical development (S1B and S1C Fig).

To further compare the RNA stability landscape between stages, we assessed the top 10% most and least stable transcripts at E11.5, E14.5, and E16.5. We observed a high degree of similarity between the 3 stages. Among the most stable transcripts, 456 were shared among all 3 stages (Fig 1D). Similarly, 491 of the least stable transcripts were shared across all 3 stages (Fig 1E). Notably, more transcripts were shared between all 3 stages than were unique to each stage. Gene ontology (GO) analysis of the most and least stable transcripts at each stage revealed consistently enriched terms shared across developmental stages (S1D and S1E Fig). Stable transcripts were enriched for functions such as ATP synthesis and protein synthesis, while unstable genes encoded regulators of transcription and cell fate determination. Importantly, these functional classes for stable and unstable transcripts are similar to those observed in other cell types in diverse organisms [36,38–41].

We next sought to identify *cis*-acting factors that shape RNA turnover dynamics in cortical cells. We first focused on 5′ and 3′ UTRs as they are known to contain regulatory elements important for RNA stability [42,43]. We examined 3′ UTR sequences of the same top 10% most stable and least stable transcripts at each stage [44]. The most stable RNAs had shorter, more GC-rich, 3′ UTRs (S1F and S1G Fig). Analysis of 5′ UTR sequences revealed a similar relationship between length and stability, with stable transcripts having shorter 5′ UTRs (S1H Fig). However, no relationship was observed between 5′ UTR GC content and stability (S1I Fig). The preferential impact of 3′ UTR GC content on RNA stability may reflect the enrichment of regulatory elements within these regions [45].

In addition to regulatory elements within UTRs, codon content within coding regions has also been implicated in mRNA stability regulation in multiple biological contexts [46–51]. To determine whether mRNA stability in the developing mouse cortex is associated with codon content, we calculated the codon stabilization coefficient (CSC), a measure of the correlation between codon usage and mRNA half-life [50]. We found robust correlations between codon usage and mRNA stability across all 3 stages, with CSC values ranging from 0.17 (stabilizing/optimal) to −0.2 (destabilizing/non-optimal) (Fig 1F). CSC values for each codon were virtually identical between E11.5, E14.5, and E16.5 (S1J and S1K Fig). To further assess the impact of codon content on mRNA stability, we binned mRNAs into quartiles based on their optimal codon content and then compared the half-lives of each quartile. We found a clear and consistent trend across all 3 developmental stages showing that mRNAs with a higher proportion of optimal codons are more stable (Fig 1G).

Finally, we examined the impact of m$^6$A, an abundant epitranscriptomic mark associated with multiple aspects of mRNA metabolism, including mRNA stability [52]. m$^6$A was previously shown to be present in >2,000 transcripts in E14.5 mouse primary cortical cells, where it was implicated in RNA degradation [19]. To assess the destabilizing impact of m$^6$A in our data set and to determine whether its role changes across development, we compared half-lives of mRNAs predicted to contain m$^6$A to those without m$^6$A at each stage [19] (Fig 1H). This analysis showed significantly lower half-lives for m$^6$A-containing transcripts, with similar extents

of destabilization at all stages. This strongly implicates m⁶A in RNA stability throughout neurogenesis, reinforcing previous findings [19]. Collectively, our data highlight multiple *cis*-acting features that shape RNA stability in cortical cells.

Although we did not observe widespread shifts in RNA half-lives across development, some variation among individual transcripts was observed (S1B and S1C Fig). We therefore aimed to identify transcripts with developmentally regulated RNA half-lives. For this, we employed a regression-based method [53] to assess RNA degradation kinetics across time in our SLAM-seq data. We first filtered the data to include only the 9,490 transcripts expressed at all 3 stages, then converted half-lives to z-scores to normalize the data between stages and allow for direct comparisons. Regression analysis was then used to identify transcripts with significant changes in stability across development. Consistent with our previous observations, half-lives for most transcripts were unchanged. However, a small subset of 294 transcripts (3.1%) had significant differences in stability across all 3 stages assayed. Hierarchical clustering of this subset based on their half-life dynamics identified 7 distinct clusters (Fig 2A and S2 Table). The temporal half-life profile for each cluster was unique (Fig 2B). Stability of cluster 2 and 3 RNAs gradually increased, while the stability of cluster 5 and 7 RNAs gradually decreased. Cluster 1 and cluster 4 stability was greatest at E16.5 and E14.5, respectively, while cluster 6 transcripts were least stable at E14.5. Altogether, these data demonstrate that turnover rates for individual transcripts are on average constant across cortical development but are dynamically regulated for a subset of transcripts.

## RNA stability correlates with developmental gene expression changes

Neural progenitors dynamically remodel their transcriptomes over developmental time and upon differentiation [7]. Yet, the extent to which RNA stability influences these transcriptional dynamics is largely unknown. To investigate this question, we assessed the stability of RNAs whose expression significantly changes across development. Because SLAM-seq is conducted using standard RNA-seq, our data set measures RNA expression levels at each stage, in addition to their half-lives. Using this data, we evaluated differential expression across the 3 stages. Using cutoffs of fold change $>2$ and $p_{adj} < 0.05$, we identified 1,056 up-regulated and 997 down-regulated transcripts at E14.5 compared to E11.5 (Fig 3A and S3 Table). Transcripts that were up-regulated from E11.5 to E14.5 included neuronal markers (e.g., *Neurod2*, *Satb2*, *Dcx*), while down-regulated transcripts included progenitor markers (e.g., *Sox2*, *Nes*, *Hes5*). These gene expression changes may reflect the increased abundance of neurons at later stages of development. We further identified 962 up-regulated and 730 down-regulated genes at E16.5 compared to E14.5 (Fig 3B and S3 Table). Similar to changes from E11.5 to E14.5, we observed increased expression at E16.5 of neuronal markers (e.g., *Satb2*, *Cux2*, *Pou3f3*/*Brn1*) and decreased expression of progenitor and cell cycle markers (*Ccnd2*, *Sox2*, *Btg2*). Notably, these data are consistent with transcriptional dynamics that are expected to occur in vivo during cortical development.

To determine whether there is a relationship between RNA stability and gene expression dynamics across development, we compared half-lives of differentially expressed transcripts. We found that RNAs whose expression is significantly up-regulated at E14.5 compared to E11.5 were marginally, but significantly, more stable compared to those that were down-regulated (Fig 3C). The same trend was observed for RNAs that were differentially expressed at E16.5 compared to E14.5 (Fig 3D). These data suggest that expression dynamics and RNA stability are coordinated. Developmentally down-regulated transcripts undergo rapid turnover, while up-regulated mRNAs are degraded more slowly to facilitate their accumulation.

Some of these developmentally regulated transcripts are known to have cell-specific expression. Given this, we investigated whether this held true across more of the transcriptome. For

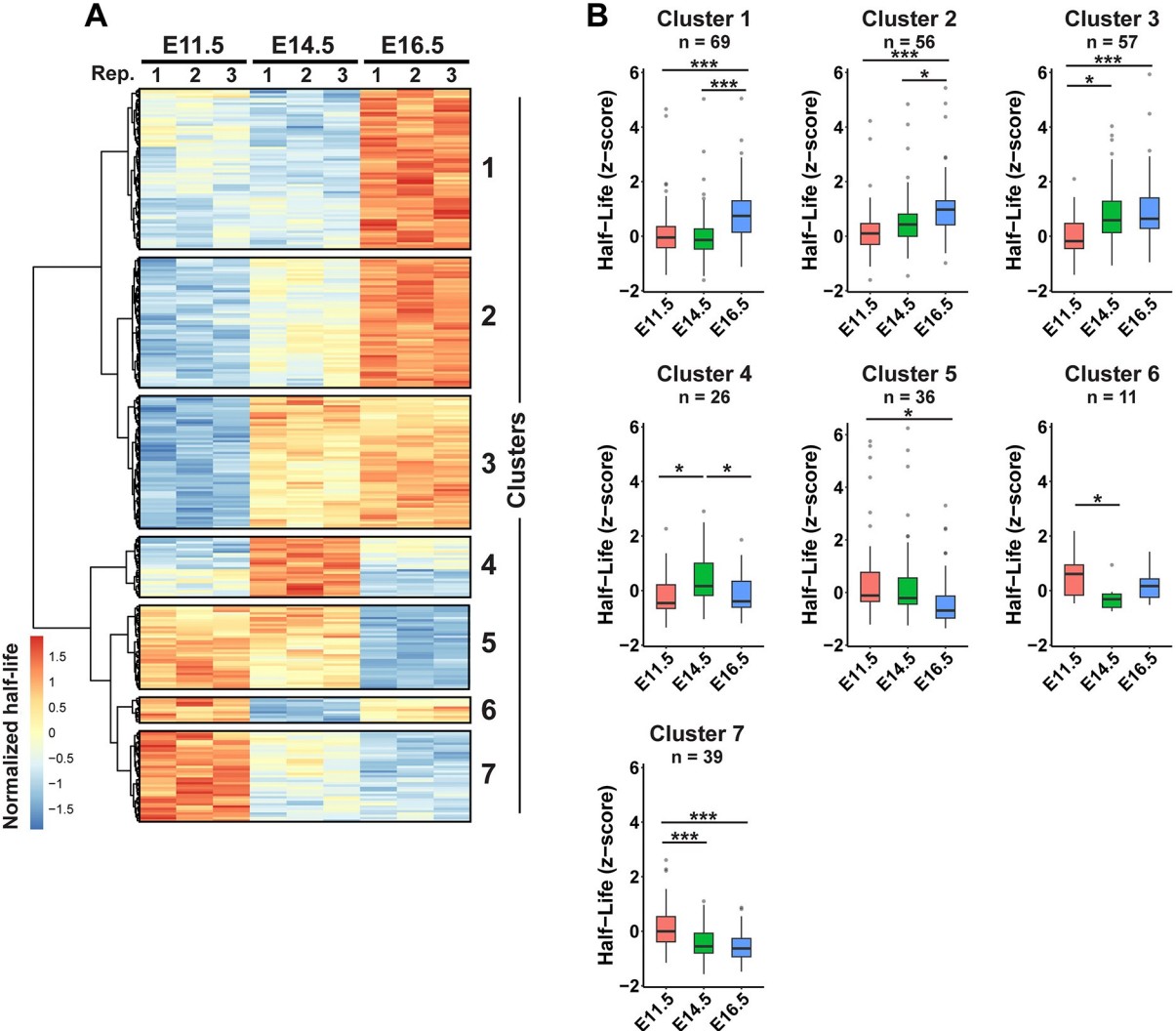

**Fig 2. Dynamic patterns of mRNA half-lives across cortical development.** (A) Heatmap showing z-score normalized half-lives for 294 significantly changing transcripts at E11.5, E14.5, and E16.5 ($n = 3$ biological replicates per stage). Numbers to the right indicate cluster labels. (B) Z-score normalized half-lives for transcripts found in the indicated clusters. *$p < 0.05$, ***$p < 0.001$. One-way ANOVA with Tukey's HSD post hoc test. Underlying data for this figure can be found in S2 Data.

this, we mined a previously published scRNA-seq data set of the developing cortex [6] to assess expression levels of developmentally regulated transcripts in specific cell types. At both transitions (E11.5 to E14.5 and E14.5 to E16.5), transcripts with decreasing expression were enriched in progenitors (RGCs and IPs), while those with increasing developmental expression were enriched in neurons (S2A and S2B Fig). Together, this suggests that gene expression differences in our data sets are linked to changes in cell composition across development.

Our data demonstrate that some developmentally regulated transcripts exhibit biased cell-type expression profiles (S2A and S2B Fig) and distinct stability profiles (Fig 3C and 3D). Therefore, we next sought to determine whether transcripts enriched in progenitors and neurons have distinct RNA half-lives. To address this, we generated a stringent list of transcripts that are highly enriched in either progenitors ($n = 56$) or neurons ($n = 57$) across 2 independent scRNA-seq data sets [6,7] (S4 Table). In our SLAM-seq data set, these transcripts were expressed at all 3 stages, with progenitor-enriched transcripts most highly expressed at E11.5,

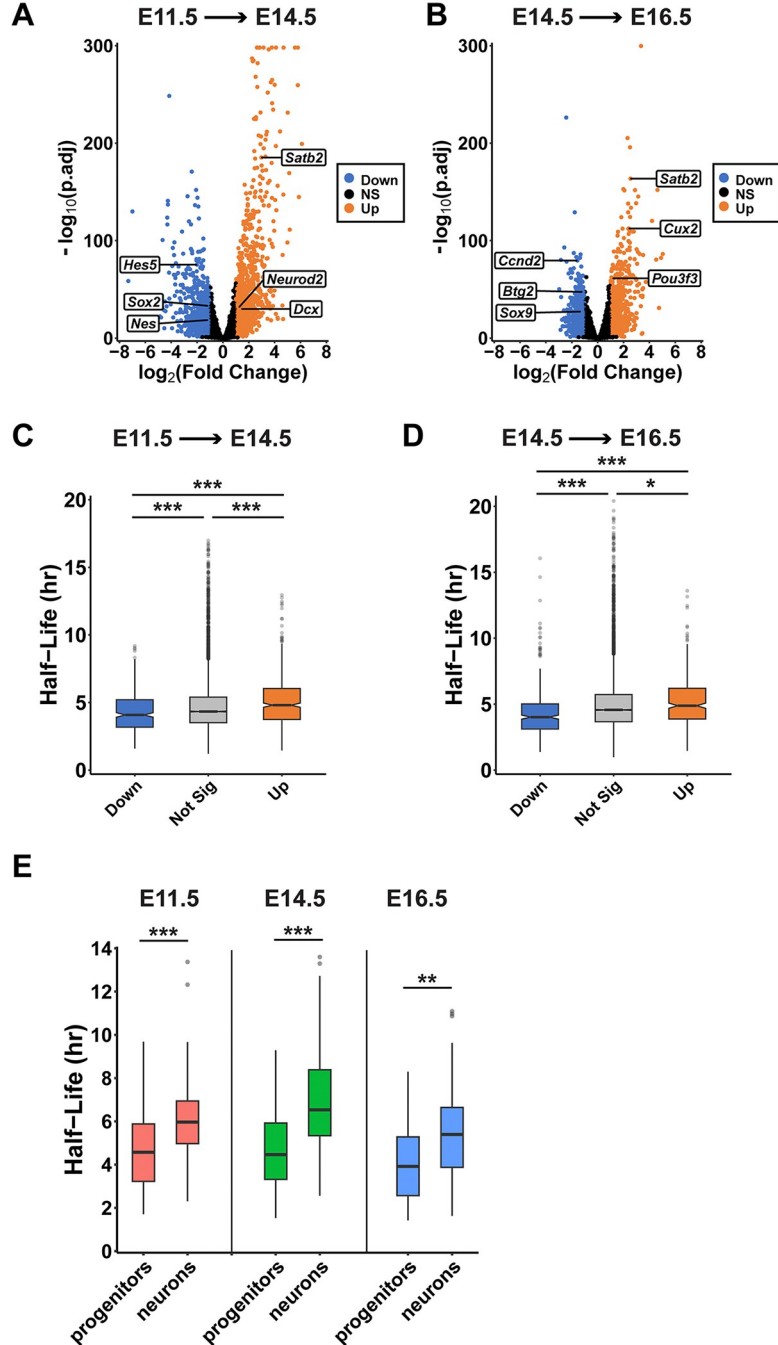

**Fig 3. Distinct stability profiles for developmentally regulated and cell type enriched mRNAs.** (A, B) Volcano plots showing transcript expression changes at E14.5 compared to E11.5 (A), or E16.5 compared to E14.5 (B). $n = 3$ biological replicates per stage. (C, D) Half-lives of differentially expressed transcripts at the indicated developmental transitions. (E) Half-lives for a subset of transcripts enriched in either progenitors ($n = 56$) or neurons ($n = 57$). **$p < 0.01$, ***$p < 0.001$. One-way ANOVA with Tukey's HSD post hoc test (C, D), Wilcoxon rank-sum test (E). Underlying data for this figure can be found in S3 Data.

and neuron-enriched transcripts most highly expressed at E16.5 (S2C and S2D Fig). These expression patterns are consistent with known cell composition at each stage. Notably, progenitor-enriched transcripts were significantly less stable than neuron-enriched transcripts

(Fig 3E). This was true across all 3 developmental stages. Collectively, these data demonstrate cell-specific layers of RNA stability during cortical development and strongly suggest that RNA stability influences developmental expression changes for a subset of transcripts. These data indicate developmental roles of RNA stability for shaping the transcriptome.

## Conditional knockout of *Cnot3* leads to microcephaly and disrupted neuronal lamination

Our data thus far indicate that RNA stability regulation is a key feature of cortical development, implicated in gene expression changes across stages. Our data also indicate that codon optimality is highly correlated with mRNA stability in the embryonic cortex (Fig 1G). Mechanistically, codon content and its effects on translation are sensed by the CCR4-NOT complex to drive degradation of non-optimal mRNAs [54]. This led us to investigate the in vivo functional consequences of mis-regulating the CCR4-NOT complex during cortical development. To address this, we focused on CNOT3, a central component of the CCR4-NOT deadenylase complex (Fig 4A). CNOT3 is essential for deadenylation [21–23] and is linked to neurodevelopmental pathologies [24–28], making it a strong candidate for investigating the role of the complex in cortical development. Importantly, inspection of scRNA-seq data sets of the developing mouse cortex revealed broad *Cnot3* expression in both progenitors and neurons [7]; however, its function in these cell types is unknown.

We generated conditional knockouts (cKOs) of *Cnot3* in RGCs and their progeny by crossing *Cnot3*^*lox/lox*^ mice with *Emx1*-Cre [55] (Fig 4B). At E12.5 (approximately 3 days after the onset of Cre-mediated recombination), we observed an 84% and 82% reduction in CNOT3 protein and RNA expression, respectively, in cKO cortices (Figs 4C and S3A). Interestingly, expression of CNOT1 was also reduced. Reduction of CNOT1 protein likely occurs via translational and/or posttranslational mechanisms, as we observed no significant change in *Cnot1* RNA measured by RT-qPCR (S3A Fig). No statistically significant changes were observed for CNOT2 expression at either the protein or RNA levels. As CNOT1 is essential for scaffolding the complex, this suggests that the integrity of the entire CCR4-NOT complex may be impaired in this mouse model. These data demonstrate the *Cnot3* cKO can be faithfully used to assess roles of this complex in cortical development.

To determine the requirement of *Cnot3* for cortical development, we first examined brains at E18.5, when neurogenesis is largely complete. cKO mice exhibited profound microcephaly, with a 38% reduction in cortical area, and a similar 38% reduction in cortical thickness (Fig 4D). In contrast, conditional heterozygous (cHet) brain sizes were comparable to control. This indicates that CNOT3 is required for overall cortical size. The cortex forms a six-layered laminar architecture, with early born neurons present in deep layers and later born neurons primarily present in superficial layers [3]. To examine the extent to which *Cnot3* ablation affects glutamatergic projection neurons, we performed immunostaining for TBR1 (Layer VI), CTIP2 (Layer V), RORβ (Layer IV), and LHX2 (Layer II/III). In cKO brains compared to control, we observed a ~28% reduction in TBR1+ neuron density, a near complete loss of CTIP2+ and RORβ+ neurons, and a 65% reduction in LHX2+ neurons (Fig 4E–4H). cHet brains were similar to control. In addition to reduced density, neuronal distribution for some markers was also disrupted in cKO brains (S3B–S3E Fig). These data indicate that CNOT3 is essential for number and laminar position of glutamatergic neurons produced throughout cortical development.

## *Cnot3* is required for viability of both progenitors and neurons

To understand how *Cnot3* controls brain size and neuron number, we next examined its requirements for neurogenesis. At E14.5, mid-neurogenesis, we observed an 18% reduction in

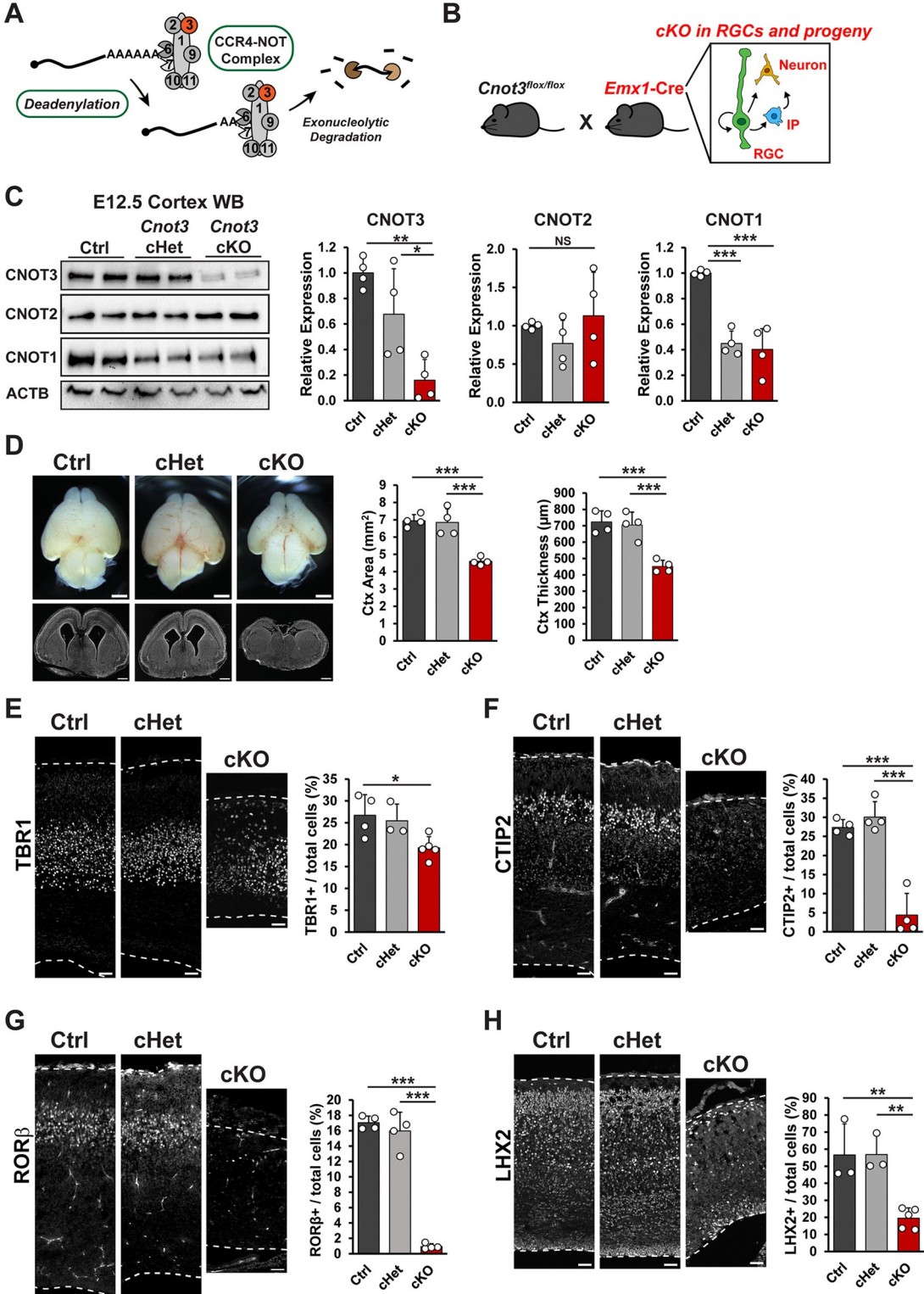

**Fig 4. *Cnot3* controls cerebral cortex size and neuronal number.** (A) Schematic of CCR4-NOT complex with CNOT3 indicated in red. (B) cKO strategy using *Emx1*-Cre to target RGCs and their progeny, including IPs and neurons. (C) Western blot using whole cortex protein lysates from E12.5 control (ctrl), conditional heterozygous (cHet), or cKO embryos. Quantifications shown to the right (*n* = 4 embryos per genotype). (D) Whole mount images and coronal cross sections of E18.5 brains. Quantifications of cortex area and thickness shown to the right (*n* = 4 embryos per genotype). (E–H)

Immunofluorescence for indicated neuronal markers in E18.5 cortical sections, with quantifications shown to the right. $n$ = 3–5 embryos per genotype. $*p < 0.05$, $**p < 0.01$, $***p < 0.001$. One-way ANOVA with Tukey's HSD post hoc test. Error bars represent standard deviation. Scale bars: 1 mm (D, top), 500 μm (D, bottom), 50 μm (E–H). Underlying data for this figure can be found in S4 Data. cKO, conditional knockout; IP, intermediate progenitor; RGC, radial glial cell.

cortical thickness in *Cnot3* cKO mice, indicating a role for *Cnot3* in development at earlier stages (Fig 5A and 5B). We performed immunofluorescence against SOX2 and PAX6 to assess RGCs. Both markers were qualitatively reduced in the proliferative zones (S4A Fig). Indeed, we observed a 26% reduction in apical SOX2+ RGCs (Fig 5C). We also assessed IPs using TBR2 expression. This showed a 43% reduction in IPs in cKO cortices compared to control (Fig 5D). This demonstrates that *Cnot3* is critical for proper number of both RGCs and IPs.

The reduction in cortical thickness and decreased progenitor and neuronal cell density could be associated with alterations in proliferation or excessive cell death. We first assessed the latter possibility by measuring apoptosis. Staining for cleaved caspase 3 (CC3) revealed minimal cell death in control, cHet, and cKO cortices at E11.5 (Fig 5E and 5F). However, beginning at E12.5, we observed widespread cell death that was predominant in cKO cortices (Fig 5E and 5F). Apoptotic cells were present throughout the cortex and were both TUJ1+ and TUJ1- (Fig 5G and 5H). This indicates that *Cnot3* loss induces apoptosis of both newborn neurons and presumed progenitors.

We next sought to determine whether neuronal cell death was strictly a result of *Cnot3* loss in progenitors, or if *Cnot3* was also required in neurons for viability. For this we crossed *Cnot3*$^{lox/lox}$ mice with *Nex*-Cre (S4B Fig), which is active in post-mitotic glutamatergic neurons beginning at E12.5 [56]. Staining for CC3 at E18.5 indicated apoptosis in *Nex*-Cre cKO cortices (S4C Fig). This indicates that neuronal cell death is due, at least in part, to an intrinsic requirement of *Cnot3* for newborn neuron viability. Staining for TBR1, CTIP2, RORβ, and LHX2 at E18.5 revealed reductions of 33%, 20%, 18%, and 10%, respectively, with only the deep layer neurons being significant (S4D–S4H Fig). The attenuated impact on superficial layers is most likely due to the later birthdate of these neurons and the time it takes for CNOT3 protein to be depleted after Cre-mediated recombination. Collectively, these data underscore the intrinsic requirement of CNOT3 for viability of multiple cell types of the developing cortex, including both neural progenitors and post mitotic neurons.

## Rescue of apoptosis partially rescues *Cnot3* cKO phenotypes

Given that *Cnot3* is essential for cell viability, we next assessed the extent to which apoptosis contributes to the loss of both progenitors and neurons in *Emx1* cKO brains. Cell death in the cortex is often associated with up-regulation of p53 signaling [10,57–59]. Consistent with this, we observed accumulation of p53 protein in E12.5 cKO cortices (Fig 6A), indicating active p53 signaling at this stage. Given this, we generated *Emx1*-Cre;*Trp53*$^{lox/lox}$;*Cnot3*$^{lox/lox}$ double cKO (dcKO) mice to abrogate p53-dependent apoptosis. Rescue of p53-dependent apoptosis was confirmed by the quantification of p53 protein accumulation and CC3+ cells in E12.5 dcKO cortices (S5A and S5B Fig). Examination of dcKO brains at E18.5 revealed a striking rescue of cortical size (Fig 6B and 6C). Notably, cortical thickness in dcKO brains was virtually unchanged compared to cKO (Fig 6D). The density of early born, deep layer neurons marked by TBR1 and CTIP2 was similar between dcKO and cKO mice (Fig 6E and 6F). In contrast, the density of later born neurons marked by RORβ and LHX2 was restored in dcKO mice (Fig 6G and 6H). This suggests that p53-dependent apoptosis explains the loss of late-born, but not early-born neurons in *Cnot3* cKO mutants. As with density, the compound mutants also failed to rescue lamination patterns for deep layer markers, but did rescue lamination of superficial

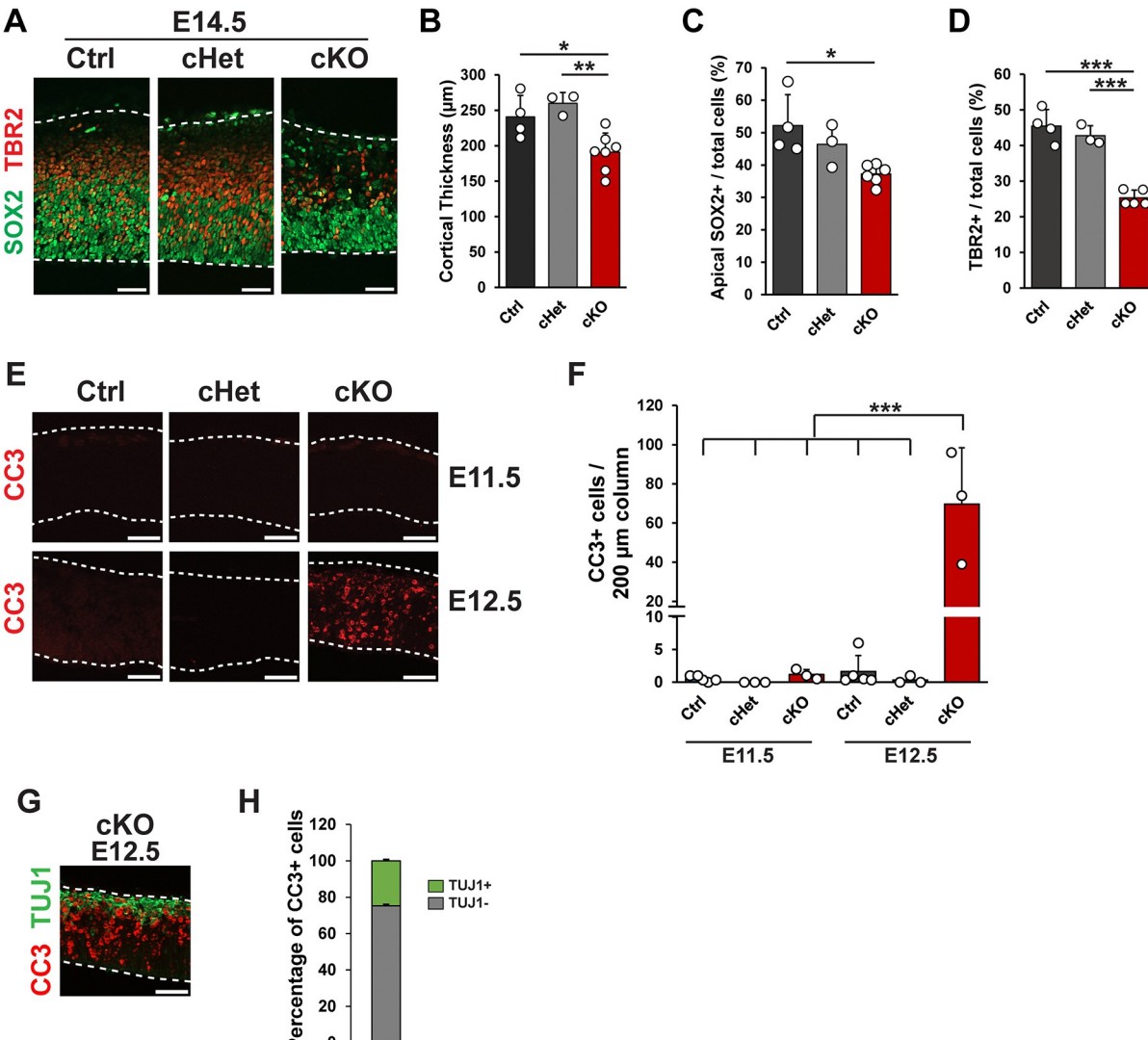

**Fig 5. *Cnot3* is required for progenitor cell number and cell survival.** (A) Immunofluorescence for SOX2 and TBR2 in E14.5 cortical sections, with quantifications shown in (B–D). *n* = 3–6 embryos per genotype. (E) Immunofluorescence for CC3 in E11.5 (top) and E12.5 (bottom) cortical sections. (F) Quantification of CC3+ cells from genotypes in (E) (*n* = 3–5 embryos per condition). (G) Immunofluorescence for CC3 and TUJ1 in an E12.5 cKO cortical section. (H) Quantification of percentage of CC3+ cells that were either TUJ1+ or TUJ1- in E12.5 cKO cortices (*n* = 3 embryos). **p* < 0.05, ***p* < 0.01, ****p* < 0.001. One-way ANOVA with Tukey's HSD post hoc test. Error bars represent standard deviation. Scale bars: 50 μm. Underlying data for this figure can be found in S5 Data. cKO, conditional knockout; CC3, cleaved caspase 3.

layers markers (S5C–S5F Fig). These data indicate both p53-dependent and p53-independent mechanisms for CNOT3 regulation of neuronal populations.

The reduction in superficial layer neurons may be due to apoptosis of progenitors at stages when these neurons are being produced. To probe this possibility, we quantified neurogenesis of compound mutant brains at E14.5. Similar to E18.5, cortical thickness at E14.5 was unchanged in dcKO brains compared to cKO (Fig 6I and 6J). However, the densities of apical SOX2+ RGCs and TBR2+ IPs were rescued in dcKO brains (Fig 6K and 6L). Thus, the decrease in progenitor density in E14.5 *Emx1*-cKO brains is due to p53-dependent apoptosis. This observation is consistent with the rescue of upper layer neurons, whose peak production

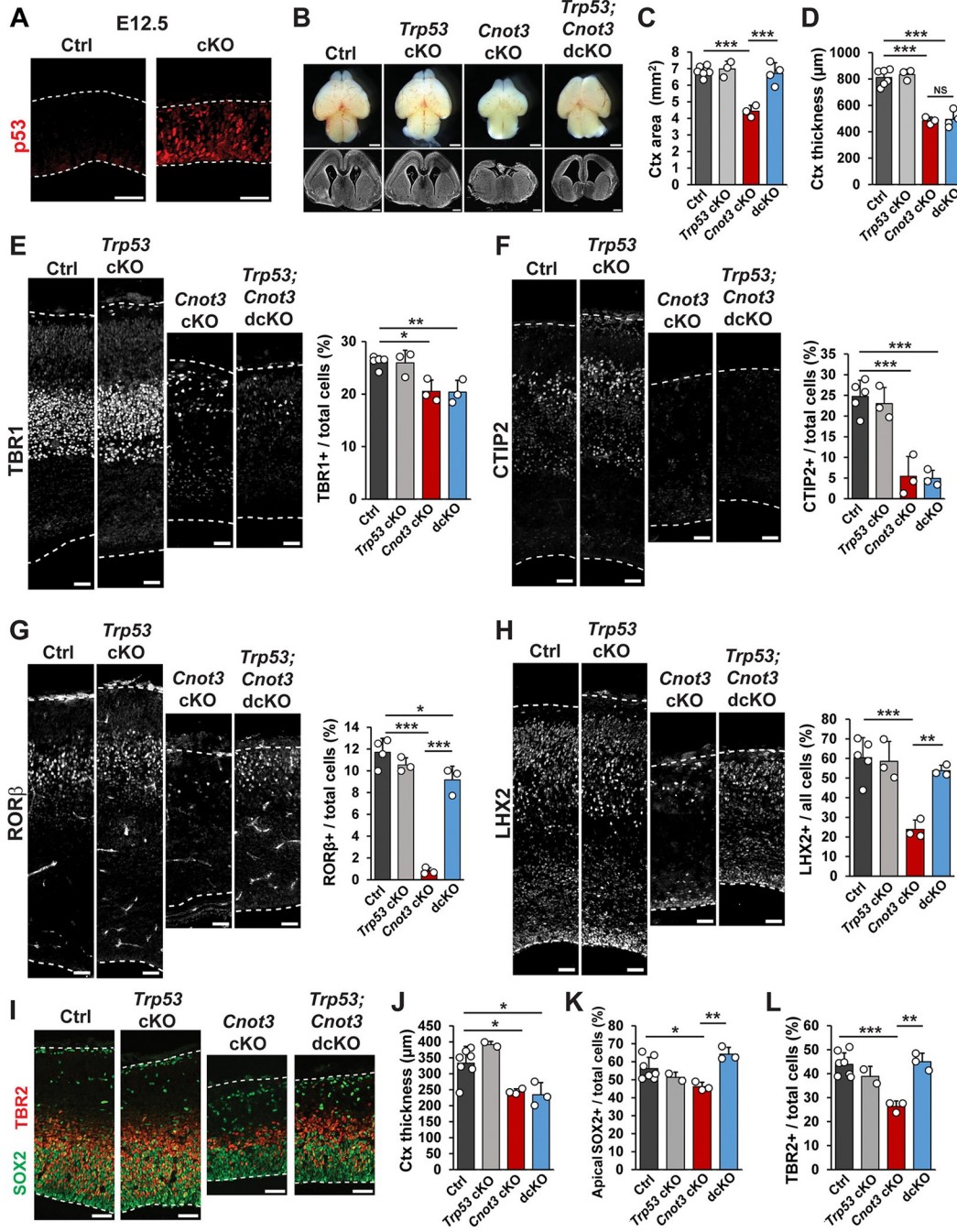

**Fig 6. p53-dependent apoptosis controls upper but not deep layer neurons.** (A) Immunofluorescence for p53 in E12.5 cortical sections. (B) Whole mount images and coronal cross sections of E18.5 brains. (C, D) Quantifications of cortex area and thickness ($n$ = 3–6 embryos per genotype). (E–H) Immunofluorescence for indicated neuronal markers in E18.5 cortical sections, with quantifications shown to the right ($n$ = 3–6 embryos per genotype). (I) Immunofluorescence for SOX2 and TBR2 in E14.5 cortical sections. (J–L) Quantification of cortical thickness, SOX2, and TBR2 at E14.5 ($n$ = 2–7 embryos per genotype). *$p < 0.05$, **$p < 0.01$, ***$p < 0.001$. One-way ANOVA with Tukey's HSD post hoc test. Error bars represent standard deviation. Scale bars: 1 mm (B, top), 500 μm (B, bottom), 50 μm (A, E–I). Underlying data for this figure can be found in S6 Data.

occurs around E14.5-E15.5. Collectively, our analysis of dcKO mice suggests that *Cnot3*-dependent microcephaly and neuronal loss are due in part to massive apoptosis.

## CNOT3 is required for regulation of cell cycle duration and cell fate

Given the findings above, we next asked if *Cnot3* controls the rate at which early-stage progenitors produce neurons. For this, we employed a previously established in vivo semi-cumulative labeling strategy to measure total cell cycle (Tc) and S-phase (Ts) durations at E12.5 [60]. BrdU was intraperitoneally injected into pregnant dams to label DNA in S-phase progenitors, followed by a second injection with EdU 1.5 h later (Fig 7A). Embryonic brains were dissected at t = 2 h and stained for BrdU, EdU, and Ki67 (proliferative cells). Total cell cycle and S-phase in control cortices were determined to be ~10 h and ~5 h, respectively (Fig 7B). Importantly, both measurements are in close agreement with previous measurements [61,62]. In contrast, in cKO cortices, total cell cycle and S-phase durations were both significantly increased to ~14 h and ~6.5 h, respectively (Fig 7B). The ratio of Ts/Tc was unchanged across the 3 genotypes, indicating that cell cycle lengthening was not biased towards S-phase (Fig 7C). These data demonstrate that *Cnot3* loss from early-stage progenitors is associated with overall slower cell cycle.

We next investigated the extent to which *Cnot3* controls early stage progenitors' ability to produce neurons. We first measured if the reduced number of early stage progenitors is due to cell death using *p53* mutants. In dcKO brains, we observed rescue of proliferative cell number, suggesting p53 contributes to E12.5 RGC number (S5G and S5H Fig). Notably, in contrast to this phenotype, deep layer neurons are not rescued by p53 (Fig 6E and 6F). Given this, we tested the possibility that CNOT3 controls fate of newborn neurons at early stages when these neurons are being generated. For this, we performed a live imaging assay to monitor the fate of dividing E12.5 progenitors and their progeny (Fig 7D and S1 and S2 Movies). Primary cell cultures were prepared from E12.5 control, cHet, and cKO cortices, and progenitors were imaged every 10 min for a period of 20 h, as previously [63,64]. After 20 h, cells were fixed and stained for SOX2 (RGCs), TBR2 (IPs), and TUJ1 (neurons) to assign the fate of progeny. In control progenitors, approximately 60% of divisions were proliferative (2 SOX2+ or TBR2+ progeny), approximately 10% were asymmetric neurogenic (one SOX2+ or TBR2+, and one TUJ1+ progeny), and approximately 30% were symmetric neurogenic (two TUJ1+ progeny). In contrast, divisions of cKO progenitors were skewed towards producing more progenitors, with approximately 80%, 5%, and 15% of divisions being proliferative, asymmetric neurogenic, and symmetric neurogenic, respectively (Fig 7E). cHet progenitors had an intermediate phenotype that was not statistically significant. The observation that progenitors undergo fewer neurogenic divisions in cKO cells suggests that *Cnot3* loss from progenitors directly impairs neuronal production. This, together with prolonged cell cycle, suggests potential mechanisms to help explain the p53-independent loss of deep layer neurons.

## Up-regulation and impaired turnover of non-optimal mRNAs in cKO cortices

Next, we aimed to understand the molecular consequences of *Cnot3* loss in the developing cortex. For this, we isolated RNA from E12.5 control, *Cnot3* cKO, and *Cnot3;Trp53* dcKO cortices and performed bulk RNA-seq. We chose E12.5, the earliest onset of phenotypes in cKO embryos, with the goal of understanding primary molecular deficits. Using cutoffs of $p_{adj} <$ 0.05 and fold-change >2, we identified 1,216 transcripts that were differentially expressed in cKO cortices compared to control, and 1,164 differentially expressed transcripts in dcKO cortices (Fig 8A and S5 Table). Targets of p53 transcriptional activation, including *Sesn2*, *Eda2r*,

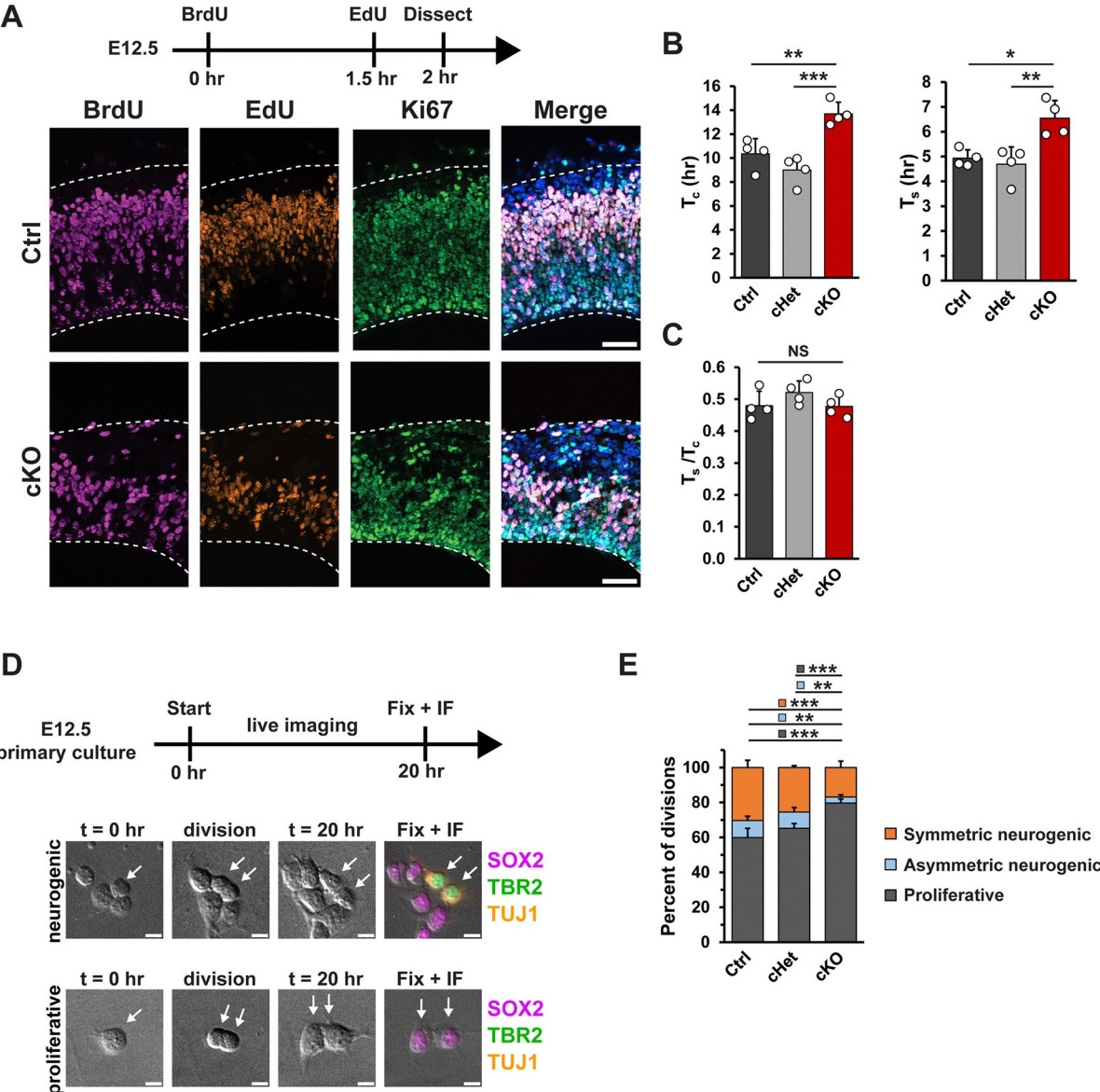

**Fig 7. *Cnot3* controls neural cell fate by modulating progenitor cell cycle duration and neurogenic divisions.** (A) Top, schematic of experimental approach for semi-cumulative labeling at E12.5. Bottom, immunofluorescence for the indicated markers in E12.5 cortical sections. (B) Quantification of total cell cycle duration ($T_c$), S-phase ($T_s$) ($n = 4$ embryos per genotype). (C) Quantification of the ratio between S-phase and total cell cycle ($T_s/T_c$). (D) Top, schematic of experimental approach for live imaging of E12.5 primary progenitor divisions. Bottom, DIC images at the beginning (t = 0 h) and end (t = 20 h) of the live imaging session and immunofluorescence after fixation for SOX2, TBR2, and TUJ1 showing representative proliferative and neurogenic divisions (n: Ctrl = 4 brains (446 divisions), cHet = 3 brains (331 divisions), cKO = 3 brains (350 divisions)). (E) Quantification of cell divisions: Proliferative (2 SOX2+TUJ1-, 2TBR2 +TUJ1-, or one of each); Asymmetric neurogenic (1 SOX2+TUJ1- or TBR2+TUJ1- and 1 TUJ1+); Symmetric neurogenic (2 TUJ1+). *$p < 0.05$, **$p < 0.01$, ***$p < 0.001$. One-way ANOVA with Tukey's HSD post hoc test (B, C). Chi-squared test with Bonferroni post hoc adjusted $p$-values (E). Error bars represent standard deviation. Scale bars: 50 μm (A), 10 μm (D). Underlying data for this figure can be found in S7 Data. cKO, conditional knockout.

*Ccng1*, and *Cdkn1a*, were among the most up-regulated genes in cKO cortices (Fig 8A and 8B). In dcKO cortices, these transcripts were each expressed at levels similar to control, confirming attenuation of p53-dependent apoptosis in the compound mutant mice. These expression changes were independently validated by RT-qPCR (Fig 8B). Importantly, this data

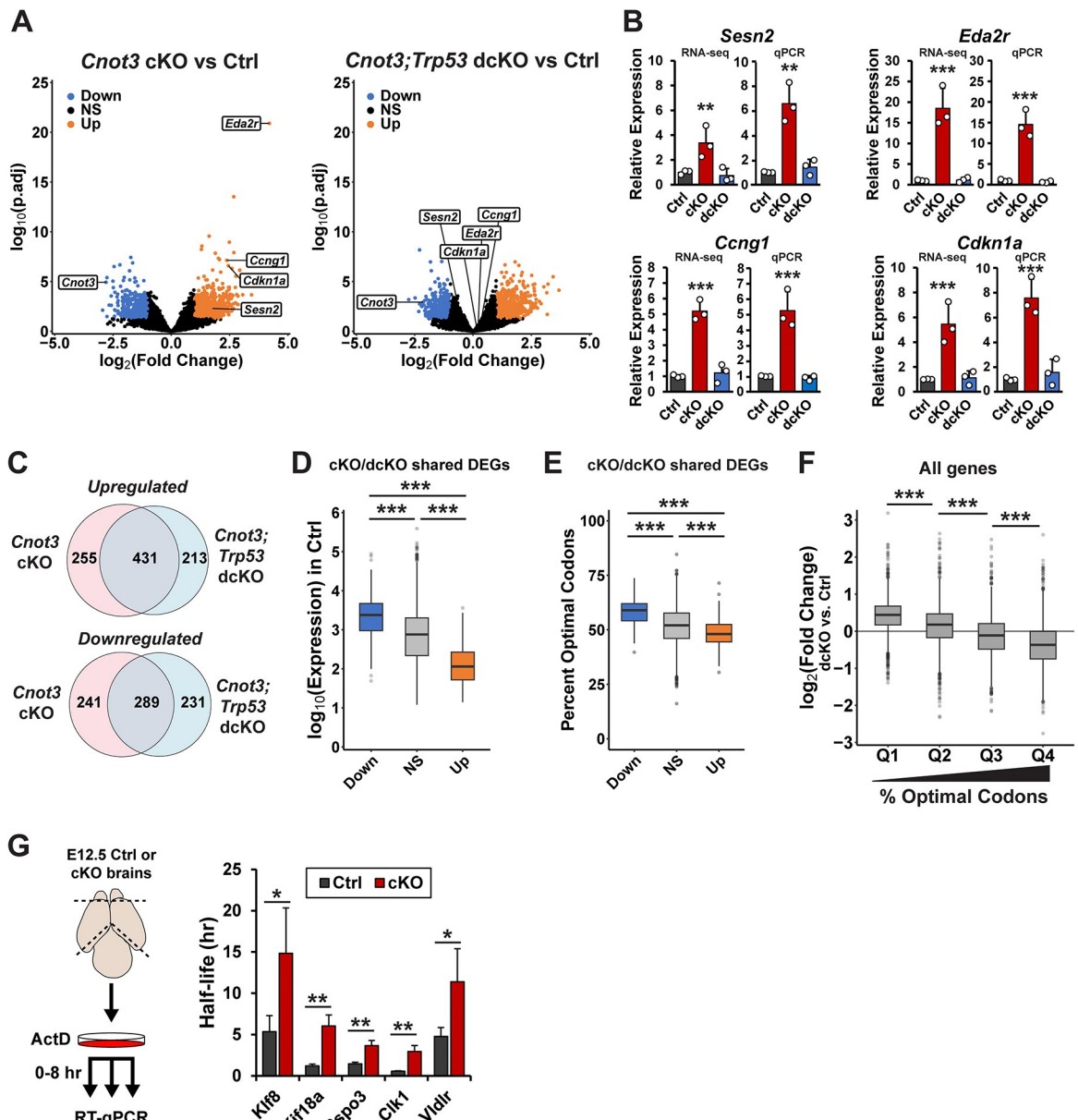

**Fig 8. Transcriptomic and stability analysis reveals molecular targets of CNOT3.** (A) Volcano plots showing differentially expressed transcripts in cKO (left) and dcKO (right) cortices versus control at E12.5 ($n = 3$ embryos per genotype). (B) Expression of p53 target genes as measured by RNA-seq (left) or qPCR (right). (C) Venn diagram showing overlap between differentially expressed genes in cKO and dcKO cortices. (D) Log$_{10}$(expression) of cKO/dcKO shared up-regulated genes (Up; $n = 431$), shared down-regulated genes (Down, $n = 289$), and all other genes (NS, $n = 13,089$) in Ctrl cortices. (E) Percentage of optimal codons (CSC > 0) for shared differentially expressed genes. (F) Log$_2$(fold change) in dcKO versus control cortices for all genes, binned into quartiles based on percentage of optimal codons. (G) Left, schematic of experimental approach for transcriptional shutoff of E12.5 primary cultures with actinomycin D (Act D). Right, half-lives of the indicated transcripts determined by RT-qPCR ($n = 4$ embryos per genotype). $*p < 0.05$, $**p < 0.01$, $***p < 0.001$. Adjusted $p$-values from DESeq2 (B (RNA-seq)). One-way ANOVA with Tukey's HSD post hoc test (B (qPCR), D–F), Welch's two-sample $t$ test (G). Error bars represent standard deviation. Underlying data for this Figure can be found in S8 Data. cKO, conditional knockout; CSC, codon stabilization coefficient; dcKO, double conditional knockout.

provides additional molecular support for p53-dependent apoptosis as a major mechanism of *Cnot3* cKO phenotypes. Given the altered cell cycle duration of progenitors in cKO cortices (Fig 7A–7C), we also investigated expression levels of transcripts involved in cell cycle

regulation. Inspection of RNA-seq data revealed that *Ccnd1*, *Ccnd2*, and *Cdk4*, were down-regulated in cKO and dcKO cortices (S6A Fig). These data provide molecular support for cell cycle phenotypes reported above and suggest potential underlying mechanisms.

Among the 686 transcripts that were up-regulated in cKO cortices, 431 (63%) were also up-regulated in dcKO cortices (Fig 8C). A similar overlap was observed for down-regulated transcripts, with 289 in common out of the 530 down-regulated transcripts in cKO cortices (55%). We focused our subsequent analysis on these shared differentially expressed transcripts as they are most likely to be regulated by *Cnot3* in a manner independent of apoptosis. Comparing our data with previously published scRNA-seq data [6] showed that differentially expressed transcripts were not biased for expression in either progenitors or neurons (S6B and S6C Fig).

To further define the molecular features of these shared *Cnot3*-dependent transcripts, we assessed their expression levels in control cortices. Up-regulated transcripts were expressed at significantly lower levels in control, compared to *Cnot3*-independent transcripts (Fig 8D). In comparison, down-regulated transcripts had significantly higher levels of expression in control brains (Fig 8D). Taken together, these data suggest that CNOT3 plays a role in helping to suppress a subset of transcripts that are typically expressed at low levels across multiple cell types.

We next sought to understand the mechanisms by which *Cnot3*-dependent transcripts are targeted. Previous work has shown that the CCR4-NOT complex is recruited to mRNAs with low codon optimality in a manner that is dependent on CNOT3 interaction with the ribosome E-site, and deletion of the yeast homolog of CNOT3 (Not5) preferentially stabilizes non-optimal mRNAs [54,65]. We thus evaluated the codon content of *Cnot3*-sensitive transcripts using CSC values calculated from our SLAM-seq data in WT cells (Fig 1H). Comparing optimal codon content among differentially expressed transcripts showed that up-regulated transcripts were significantly less optimal than unchanged transcripts, while down-regulated transcripts were more optimal (Fig 8E). This data is consistent with preferential targeting of non-optimal RNAs for degradation by the CCR4-NOT complex. We next asked whether this phenomenon could be observed transcriptome-wide or if it was specific to transcripts that were significantly altered in mutant cortices. Binning all mRNAs into quartiles based on percentage of optimal codons revealed that the most non-optimal mRNAs tended to be the most up-regulated in *Cnot3* mutants (Fig 8F). Collectively, these data implicate *Cnot3* in regulating RNA expression in a manner dependent on codon content, with preferential up-regulation of non-optimal mRNAs in mutant cortices. This further supports RNA stability control as a mechanism by which *Cnot3* regulates cortical development.

Disruption of the CCR4-NOT complex in *Cnot3* cKO cortices is expected to impair deadenylation and stabilize target transcripts. To evaluate whether increased expression of non-optimal mRNAs is due to changes in mRNA stability, we measured mRNA half-lives in primary cortical cells using transcriptional shutoff with Actinomycin D (ActD). Primary cultures were generated from E12.5 control and cKO cortices and cells were harvested 0, 2, and 8 h after addition of ActD to assess RNA degradation by RT-qPCR (Fig 8G). We examined the decay kinetics of 5 mRNAs whose expression increased in cKO cortices (S6D Fig) and that fell into the lowest quartile of codon optimality. All 5 transcripts were stabilized in cKO cells, consistent with defective mRNA turnover in the absence of *Cnot3* (Figs 8G and S6E). Of note, three of these genes (*Klf8*, *Kif18a*, and *Clk1*) have roles in cell cycle regulation [66–68]. A fourth gene (*Vldlr*) is a reelin receptor important for cortical neuron positioning [69]. In sum, these data reveal up-regulation of non-optimal, poorly expressed RNAs in cKO cortices. Transcriptional shut-off analysis of select transcripts confirms impaired RNA degradation in the absence of *Cnot3*. These observations directly implicate the CCR4-NOT complex in controlling gene expression in the developing cortex.

## Discussion

RNA regulation in the developing cortex is dynamic, with rapid changes in expression across dual axes of time and differentiation. RNA expression levels are determined by complementary rates of synthesis and degradation, but the quantitative contribution of the latter to cortical development is largely unknown. We apply omics analyses and genetic approaches to define how RNA turnover controls cortical development. Our transcriptome-wide survey of RNA half-lives across development reveals an in-depth understanding of the *cis* factors that contribute to RNA turnover, as well as previously unappreciated relationships between RNA stability and developmental gene expression changes. Genetic manipulation of the CCR4-NOT complex demonstrates the consequences of mis-regulating RNA turnover in vivo, indicating that CNOT3 is critical for neurogenesis. Our findings reinforce essential roles for RNA stability regulation in cortical development, highlighting new mechanisms relevant for related neuropathologies.

### A new landscape of mRNA stability in the developing cortex

Our data define the landscape of RNA degradation across developmental time. We provide evidence for 2 distinct modes of RNA stability regulation in the cortex. First, a dynamic half-life model in which the stability of a given RNA varies across development. Second, a static half-life model in which RNA degradation rates are constant across development. While these models are not mutually exclusive, our data indicate that a static half-life model is predominant in the developing cortex (Fig 9A), as RNA decay kinetics for most transcripts are unchanged across development. While dynamic half-life regulation occurs for a subset of transcripts (Fig 2), this appears to be the exception rather than the rule.

We discovered an unexpected relationship between RNA half-life and expression changes across time. Specifically, we find that transcripts that increase in expression at later stages of cortical development tend to have longer half-lives, while transcripts that decrease in expression tend to have shorter half-lives (Fig 9A). We propose that transcripts required during early, but not late, developmental stages have short half-lives to help promote their rapid clearance when they are transcriptionally silenced at later stages. In contrast, the relative higher stability of transcripts needed later in development may promote their accumulation over time. Our data suggests that RNA stability acts as an additional regulatory layer to help ensure proper timing of gene expression in the developing cortex.

Our data also give some insight into the extent to which RNA half-lives vary across cortical cell types. We measured RNA half-lives as the cellular composition of the cortex transitions from progenitor-enriched at E11.5 to neuron-enriched at E16.5. If the half-life of a given transcript greatly differed between progenitors and neurons, we expect to have observed a shift in RNA stability kinetics from E11.5 to E16.5. However, we did not observe widespread changes in half-lives across these developmental transitions. Importantly, this suggests that turnover rates for most transcripts may be similar, regardless of the cell type in which they are expressed. It is important to note we do not rule out the possibility of cell type specific stability regulation for a small number of transcripts. Further, while we used scRNA-seq data sets and knowledge of cell composition changes from E11.5 to E16.5 to inform these conclusions, our study did not directly compare RNA decay in progenitors and neurons. Single cell analysis of RNA stability has successfully been employed in other complex model systems, including intestinal organoids and zebrafish embryos [70,71]. Adapting these techniques for in vivo analysis of RNA stability across diverse cell types of the developing mammalian cortex is an exciting direction for future investigation.

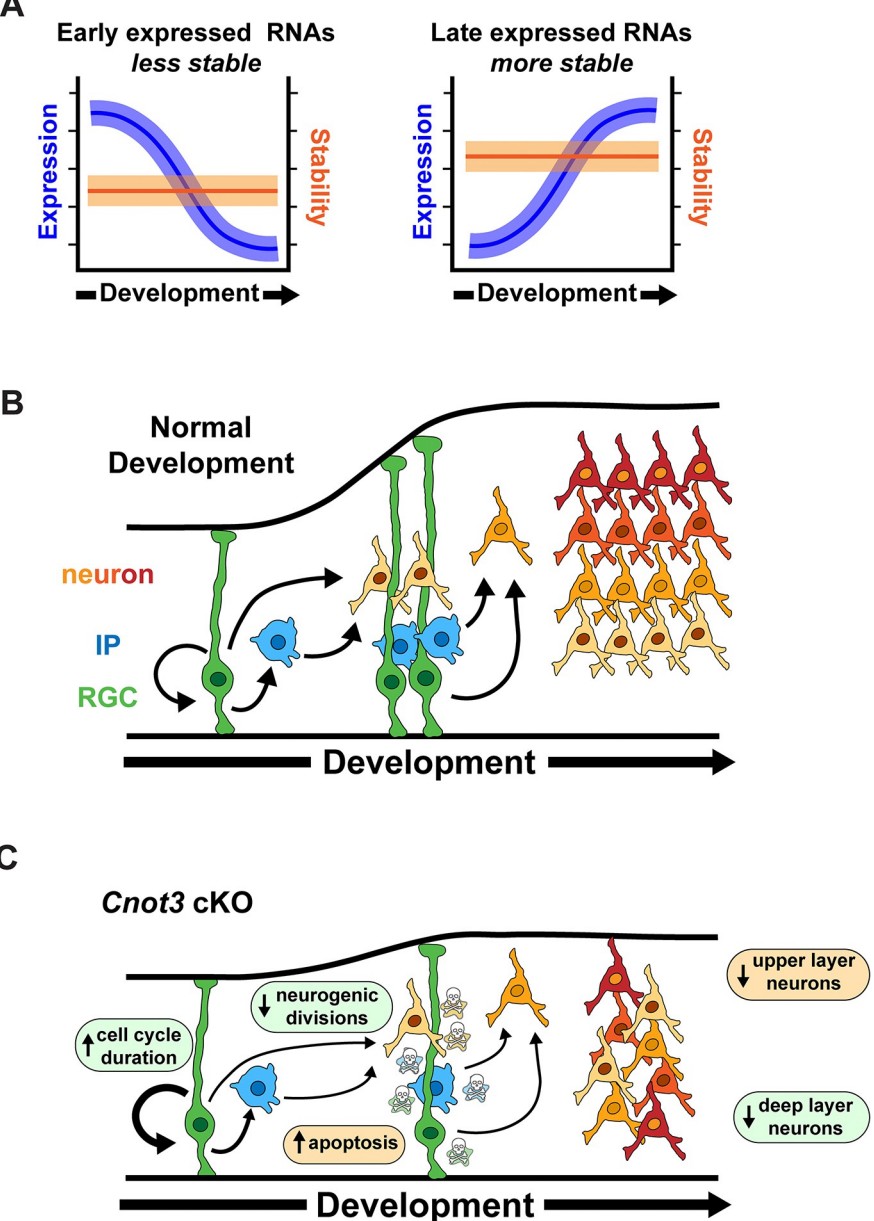

**Fig 9. Models for relationship between RNA turnover and gene expression, and the role of *Cnot3* in cortical development.** (A) Transcripts that are down-regulated across development are relatively unstable, while those that are up-regulated across development are relatively stable. For the majority of transcripts, RNA half-lives are static across development. Cortical development in WT (B) and *Cnot3* cKO embryos (C). Loss of *Cnot3* impacts development by p53-dependent and p53-independent mechanisms leading to apoptosis of progenitors and neurons, reduced neurogenic divisions, and longer cell cycle duration. Our data suggest that apoptosis is especially relevant for reduced number of upper layer neurons. cKO, conditional knockout.

## The CCR4-NOT complex is essential for cortical development

Our study used an orthogonal genetic approach to demonstrate essential roles of RNA stability in cortical development by focusing on the CCR4-NOT complex. To that end, we used progenitor and neuronal Cre drivers to demonstrate requirements of *Cnot3* in both neural progenitors and newborn neurons of the developing cortex. While we specifically targeted

CNOT3 in progenitors, expression of the central scaffold CNOT1 was also affected, suggesting the entire complex may be affected (Fig 4C). *Cnot3* cKO brains are microcephalic, with fewer progenitors and neurons. We show these phenotypes are due in part to widespread p53-dependent apoptosis of both populations. Interestingly, *Cnot3* cKO does not induce apoptosis in mouse ESCs or male germ cells [33,35], but does in B cells by directly regulating the stability of *p53* mRNA [72]. These observations collectively suggest that the requirement of *Cnot3* for mammalian cell viability may be cell type specific.

Using compound mutants, we mechanistically define how *Cnot3* controls cortical neurogenesis, in both a p53-dependent and p53-independent fashion (Fig 9B and 9C). By analyzing cKO and dcKO brains at different stages, we defined the timing of when these mechanisms are predominant. We observed a preferential rescue of superficial layer neuron density at E18.5, coupled with the rescue of progenitor density at E14.5, when superficial neurons are being born. This suggests that p53-dependent mechanisms are especially prevalent at later stages of cortical neurogenesis. In contrast, loss of early born deep layer neurons was largely p53-independent.

This raises the question of what mechanisms control deep layer neuron number. We observe fewer proliferating cells in E12.5 cKO cortices, and this phenotype was rescued in dcKO brains. However, in contrast, loss of deep layer neurons at E18.5 is not rescued by *p53*. This suggests there must be apoptosis-independent mechanisms at play. Importantly, our semi-cumulative labeling and live imaging data in cKO cells (Fig 7) demonstrate that E12.5 progenitors take longer to divide and tend to produce more progenitors at the expense of neurons. We expect that these progenitor cell fate changes ultimately lead to production of fewer deep layer neurons. How *Cnot3* controls progenitor cell cycle and neurogenic divisions is a fascinating question for future studies. It is notable that similar phenotypes have been observed for mutations in other RNA-binding proteins [73], reinforcing the role of posttranscriptional control in cell fate specification.

## CNOT3 as a regulator of gene expression in the developing cortex

To understand the molecular underpinnings of these cortical development phenotypes, we measured transcriptomic changes using RNA-seq. While the biological implications for the majority of dysregulated transcripts remain elusive, this analysis provided several key insights into potential roles of CNOT3-mediated gene regulation in the cortex. First, we observed increased expression of p53 transcriptional targets in cKO cortices, including *Eda2r*, *Sesn2*, and *Ccng1*. This data underscores p53-dependent mechanisms as a major driver of cortical defects in cKO embryos, including the reduction in superficial layer neurons. We also observed decreased expression of several cell cycle-related transcripts, including *Ccnd1*, *Ccnd2*, and *Cdk4*. These observations align with our in vivo measurements of prolonged cell cycle, as well as previous reports indicating a role for CNOT3 in cell cycle regulation [32,74,75]. Of note, *Cdk4* is also down-regulated upon knockdown of *CNOT3* in human cardiomyocytes, suggesting a conserved mechanism of cell cycle regulation [32].

Although the CCR4-NOT complex has roles in multiple aspects of RNA metabolism [20], it is primarily described as a regulator of mRNA stability. While we cannot rule out the possibility that cKO phenotypes are caused in part by disruption of other molecular processes, several lines of evidence suggest that RNA turnover is impaired in cKO cortices. First, gene expression changes in cKO and dcKO cortices correlated with codon content, with non-optimal mRNAs showing the highest up-regulation of transcript levels. In light of previously established roles of the CCR4-NOT complex in degrading non-optimal RNAs [54,76], this is consistent with widespread RNA stabilization in mutant cortices. Second, using transcriptional shutoff experiments

we directly show stabilization of non-optimal mRNAs in cKO cells. Overall, this data highlights the importance of maintaining control of RNA degradation in the developing cortex.

The CCR4-NOT complex, including CNOT3, is increasingly implicated in human disease. De novo mutations in human *CNOT3* cause severe neurodevelopmental deficiencies characterized by a spectrum of intellectual disability, speech delay, seizures, and behavioral problems [24–30]. Further *CNOT3* is a high confidence syndromic Autism gene with a SFARI score of 1S, while *CNOT1* is considered 2S. Importantly, recent work has demonstrated that many autism risk and intellectual disability genes converge on a common pathway characterized by impaired neurogenesis including asynchronous development of deep layer neurons [77,78]. Along these lines, we found aberrant neurogenesis including of deep layer neurons in *Cnot3*-deficient cortices. Together, these data suggest that alterations in cortical development may be an important contributor to neurodevelopmental pathologies associated with *CNOT3* mutation. Further study of this cKO mouse model may thus provide important insights into the etiology of *CNOT3* pathologies. Additionally, an important next step for future studies will be to investigate how germline nonsense and missense mutations of *Cnot3* impact the developing cortex.

## Materials and methods

### Mice

Animal use was approved by the Duke Institutional Animal Care and Use Committee (Protocol #: A060-22-03) and followed ethical guidelines provided by the Duke Division of Laboratory Animal Resources. Mouse lines were previously described: *Emx1*-Cre JAX stock #005628 [55], *Trp53*$^{lox/lox}$ JAX stock #008462 [79], *Nex*-Cre [56], *Cnot3*$^{lox/lox}$ [33]. Primers used for genotyping are listed in S6 Table. For embryo staging, plug dates were defined as embryonic day (E)0.5 on the morning the plug was identified.

### SLAM-seq

**Sample collection and library preparation.** Embryos were dissected in cold PBS and cortical tissue was microdissected from 3 biological replicates. Each biological replicate consisted of pooled cortical tissue from 7 embryos (E11.5) or 2 embryos (E14.5 and E16.5). Cortices were dissociated by incubation with 0.25% Trypsin (Thermo Fisher) at 37˚C for 5 min (E11.5), 10 min (E14.5), or 20 min (E16.5). Trypsinization was halted by addition of trypsin inhibitor (Sigma T6522) diluted in neural progenitor media [DMEM (Thermo Fisher 11995–065) supplemented with N-acetyl-L-cysteine (Sigma A9165, 1 mM), N2 (Thermo Fisher 17502048, 1X), B27 without vitamin A (Thermo Fisher 12587–010, 1X), and mouse bFGF (R&D Systems 3139-FB025, 10 ng/ml). Following trituration with a P1000 pipette, cells were pelleted at 200 g for 5 min to remove debris then resuspended in fresh neural progenitor media. Cells from each biological replicate were divided into 5 wells of a 24-well plate coated with poly-D-lysine (Sigma P7280) and placed at 37˚C in a humidified incubator with 5% $CO_2$. Cells were allowed to attach to the plate for 2 h, then media was replaced with fresh neural progenitor media containing 100 μm 4sU (Sigma T4509). The time of 4sU addition was designated as the t = −20 h time point. Media was replaced with fresh 4sU containing media at t = −12, −8, −4, and −1 h. At t = 0 h, a 50× molar excess (i.e., 5 mM) of uridine was added to the media. Cells from a single well per replicate were lysed 0, 2, 4, 8, and 12 h after addition of uridine by addition of RLT buffer (Qiagen RNeasy Plus Micro Kit, cat. 74034) supplemented with 1 mM DTT. Lysates were stored at −80˚C before further processing. RNA was extracted from lysates using the Qiagen RNeasy Plus Micro Kit according to manufacturer's instructions, except all buffers were supplemented with 1 mM DTT. Alkylation of 4sU RNA was performed as previously described

[36] using approximately 1.5 μg of RNA per replicate. After alkylation, RNA was re-purified using the RNA Clean and Concentrator Kit (Zymo 11–325). Sequencing libraries were generated using 400 ng of alkylated input RNA per sample and the Illumina Stranded mRNA Prep Kit (cat. 20040532) according to the manufacturer's instructions.

**Read mapping and half-life calculation.**   Sequenced reads were processed with the SlamDunk tool v0.4.3, using the mm10 genome and exons extracted from the GENCODE vM24 gene annotation. The number of covered (T)s and converted (T)s for each exon was extracted from the SlamDunk results. Covered (T) counts for all exons within a gene were summed together, and the same was done for converted (T)s. A conversion rate was calculated for each gene as the percentage of converted (T)s to covered (T)s. These conversion rates were calculated for each time point, and a half-life value was calculated for each gene based on the decay in these conversion rates over time. The decay curve was estimated in R with the "nlsLM" function from "minpack.lm" package, using the model "y ~ exp(-(k*x))", where "x" are the time points and "y" are the conversion rates scaled such that the initial conversion rate (at time point 0) is 1. The "k" value was extracted from the fit curve, and the half-life was calculated as "ln(0.5) / -k".

**Quality control.**   After calculating half-lives, the data was filtered for read coverage and the quality of half-life calculation. The following filters were applied to each transcript within each biological replicate: coverage on T > 500 at each time point of the chase, half-life < 24 h and > 0 h, and $R^2$ > 0.6. Transcripts passing these filtering steps in all 3 biological replicates per stage were retained for further analysis and are listed in S1 Table.

**GO analysis.**   Gene ontology analysis for top 10% most and least stable transcripts at each stage were performed using Panther [80].

**Codon stabilization coefficient (CSC).**   Mouse coding region sequences for were obtained from Ensembl using the biomaRt package in R [81]. For genes with multiple transcript isoforms, the longest isoform was used. Codon frequency for each transcript was calculated using the oligonucleotideFrequency function within the Biostrings [82] package using width = 3, step = 3, and as.prob = T. CSC for each codon was calculated as the Pearson correlation coefficient between codon frequency and transcript half-life. Optimal codons were defined as those with CSC > 0, and non-optimal codons were defined as those with CSC < 0.

**maSigPro.**   For statistical analysis of half-life changes across each of the 3 stages, transcript lists were filtered to include only those that were present in all 3 data sets. Half-lives were z-score normalized and transcripts with significant differences across all 3 stages were identified using the maSigPro [53] package in R. The following modifications were made from the default parameters: for the regression fitting T.fit function, alfa = 0.05 was used, to extract the significant genes using the get.siggenes function, rsq = 0.6 was used. Significant genes were then clustered using the pheatmap function in R with clustering_method = "ward.D2".

**Differential expression.**   Differential expression between stages was determined using DESeq2 (ref. [83]). Count data from the t = 0 h time point for each replicate was used to generate the input count matrix. Data was filtered to include only those genes that had >10 reads in all samples. Significantly differentially expressed genes between E11.5 and E14.5, or E14.5 and E16.5 were defined as those with |fold-change| > 2 and $p_{adj}$ < 0.05.

**Generating progenitor and neuron enriched gene lists.**   Lists of genes enriched in progenitors or neurons were generated using 2 previously published scRNA-seq data sets [6,7]. Independent enrichment lists were generated from each data set and genes that were in common between each were used in this study (S4 Table). For Telley and colleagues [7], data was accessed online (http://genebrowser.unige.ch/telagirdon/) and the "Genes with similar dynamics" feature was used to generate 3 lists of the top 100 genes most similar to *Sox2* (RGCs), *Eomes* (IPs), and *Neurod2* (neurons). The *Sox2* and *Eomes* lists were combined to create the

progenitor list for the Telley and colleagues [7] data set. For Di Bella and colleagues [6], the gene expression matrix was downloaded from the Single Cell Portal (https://singlecell. broadinstitute.org/single_cell/study/SCP1290/molecular-logic-of-cellular-diversification-in-the-mammalian-cerebral-cortex). The FindMarkers function in Seurat [84] was used to identify genes that are at least 1.5-fold enriched and expressed in at least 50% of cells of the desired cell type. Genes found to be enriched in "Apical Progenitors" and "Intermediate Progenitors" were combined to generate the progenitor-enriched list. Genes enriched in "Immature Neurons," "DL CPN," and "UL CPN" were combined to generate the neuron-enriched list.

## Western blots

E12.5 cortices were homogenized in cold RIPA buffer supplemented with Halt Protease Inhibitor (Thermo Fisher 7834). Lysates were incubated on ice for 10 min then clarified by centrifugation at 15,000 rpm for 5 min at 4˚C. For western blot, 10 μg of protein lysate was mixed with Laemmli buffer (1× final) and DTT (25 mM final) and boiled at 95˚C for 5 min. Samples were loaded into Mini-PROTEAN TGX precast gels (Bio-Rad) and run at 100 to 150V for 1 to 2 h. Transfer was performed using the Trans-Blot Turbo Transfer System (Bio-Rad). Membranes were blocked in 5% milk in TBST then incubated with primary antibodies overnight at 4˚C. The following primary antibodies were used: CNOT1 (CST 44613S, 1:1,000), CNOT2 (Thermo Fisher 10313-1-AP, 1:1,000), CNOT3 (Thermo Fisher 11135-1-AP, 1:500), ACTB (Santa Cruz sc4778, 1:500). After primary antibody incubation, blots were washed 3 times with TBST, then incubated with secondary antibody for 1 h at RT. The following secondary antibodies were used: goat-anti-mouse-HRP (1:5,000), goat-anti-rabbit-HRP (1:5,000). After secondary antibody incubation, blots were washed 3 times with TBST, then exposed to Pierce ECL Western Blotting Substrate (Thermo Fisher). Imaging was performed using the Gel Doc XR system (Bio-Rad). Quantification was performed using ImageJ.

## Immunofluorescence

Embryonic brains were fixed in 4% PFA at 4˚C overnight, washed 3 × 10 min in cold PBS, then incubated in 30% sucrose at 4˚C overnight. Brains were frozen in NEG-50 (Fisher), and 20 μm thick coronal sections were collected on charged glass slides and stored at −80˚C until staining. For IF, frozen sections from the somatosensory cortex were thawed for 5 to 10 min at RT then washed twice in PBS. For RORβ staining only, antigen retrieval was performed by boiling sections in sodium citrate buffer (10 mM sodium citrate, 0.05% Tween 20 (pH 6.0)) for 20 min, then cooling to RT for 10 min followed by 3 PBS washes. Sections were permeabilized with 0.3% TritonX-100 in PBS for 15 min and blocked in 5% NGS in PBS for 1 h at RT. Sections were incubated with primary antibodies for 2 h at RT and secondary antibodies for 30 min at RT. The following primary antibodies were used: SOX2 (Thermo Fisher 14-9811-82, 1:250), PAX6 (Millipore AB2237, 1:250), TBR2 (Abcam ab183991, 1:1,000), TUJ1 (Biolegend 801202, 1:1,000), TBR1 (CST 49661S, 1:1,000), CTIP2 (Absolute Antibody Ab00616-7.4, 1:500), RORβ (R&D Systems N7927, 1:100), LHX2 (Millipore ABE1402, 1:500), p53 (Leica CM5, 1:250), CC3 (CST 9661S, 1:250). Slides were mounted using either Vectashield (Vector Labs) or Fluoromount G (Thermo Fisher).

## Image acquisition and quantification

Images were captured using a Zeiss Axio Observer Z1 equipped with an apotome for optimal sectioning at 5× and/or 20×. For each experiment, 2 to 3 sections were imaged per brain. Images were captured with identical exposures. For image processing, maximum intensity projections were generated from Z-stacks and brightness was adjusted equivalently for all

images using FIJI software. For quantification, images were cropped to 200 μm wide radial columns and cells were counted manually using FIJI cell counter or automatically using QuPath [80]. For QuPath quantification, parameters were adjusted as follows: requested pixel size = 0.1 μm, background radius = 5 μm, minimum area = 10 μm$^2$, cell expansion = 2 μm, include cell nucleus and smoother boundaries boxes were unchecked. The threshold was set independently for each channel and maintained for each image in an experiment. For binning analysis, spatial coordinates for each detected cell were exported from either FIJI or QuPath and cells were assigned to one of 5 equally sized bins spanning from the ventricle to the pia to calculate the distribution.

## Live imaging

E12.5 cortices were dissected and dissociated as described above for SLAM-seq sample preparation with the following modifications: only 1 embryo was used per biological replicate, trypsinization was performed for 5 min, after resuspension in neural progenitor media, 175,000 cells were plated onto a poly-D-lysine treated 24-well glass-bottomed plate (MatTek). Cells were allowed to adhere to the plate for 2 h, then images were captured every 10 min for 20 h using a Zeiss Axio Observer Z1 microscope fitted with a Pecon incubation chamber as previously described [85]. After 20 h, cells were fixed in PFA for 20 min, permeabilized with 0.1% Triton-X100 for 10 min, blocked in 5% NGS for 30 min, incubated with primary antibody for 1 h, and secondary antibody for 30 min. All immunostaining steps were performed at RT. The following primary antibodies were used: SOX2 (Thermo Fisher 14-9811-82, 1:1,000), TBR2 (Abcam ab183991, 1:500), and TUJ1 (Biolegend 801202, 1:2,000).

## Semi-cumulative labeling

Pregnant dams were intraperitoneally injected with BrdU (50 mg/kg) at t = 0 h, followed by EdU (10 mg/kg) at t = 1.5 h. Embryos were dissected at t = 2 h. EdU was visualized using the Click-iT Plus Imaging Kit (Thermo Fisher C10638), followed by immunostaining using Ki67 (CST 12202, 1:250) and BrdU (Abcam ab6326, 1:200) primary antibodies as described above. Total cell cycle (Ts) and S-phase (Ts) were calculated as follows: S cells = EdU+; L cells = BrdU +EdU-; P cells = Ki67+; Ts = (S cells / L cells) * 1.5 h; Tc = (P cells / S cells) * Ts.

## Bulk RNA sequencing

E12.5 embryos were dissected in cold PBS and cortical tissue from single embryos were microdissected and flash frozen on dry ice. Three biological replicates were used per genotype. Tissue was homogenized in RLT buffer (Qiagen) and RNA was extracted using the RNeasy Plus Micro Kit (Qiagen). Libraries were prepared using the TruSeq Stranded mRNA kit (Illumina) and sequenced on the NovaSeq platform with 150 bp paired-end reads. Read quality was assessed with FastQC and adapter trimming was performed using Cutadapt. Reads were aligned to the mm39 genome using STAR [86]. Mapped reads were counted and assigned to genes using featureCounts [87], and differential expression in cKO and dcKO versus control samples was determined using DESeq2 [83].

## Transcriptional shut-off

E12.5 cortices were dissected and dissociated as described above for SLAM-seq sample preparation with the following modifications: only 1 embryo was used per biological replicate, trypsinization was performed for 5 min, after resuspension in neural progenitor media each replicate was split into 3 wells of a poly-D-lysine coated 24-well plate. At t = 0, actinomycin D

(Sigma A9415) was added to the media to a final concentration of 5 μg/ml. Cells were lysed at t = 0, 2, 8 h by addition of Trizol (Thermo Fisher 15596026) and RNA was extracted according to manufacturer's instructions. cDNA was generated from 500 ng total RNA using iScript cDNA Synthesis Kit (Bio-Rad). qPCR was performed using iTaq SYBR Green Supermix (Bio-Rad). qPCR primers are listed in S6 Table. Decay data was fitted to the single exponential decay equation (y = $y_o$ * $e^{-kt}$) using the nls function in R to determine the degradation rate constant, k. Half-life was calculated as $t_{1/2} = (\ln(2))/k$.

## Statistical analysis

Sample collection, data acquisition, and quantifications were performed blind with respect to genotype. Statistical tests, *p*-values, and sample size for each analysis are reported in S7 Table.

## Supporting information

**S1 Fig. Features of mRNA stability in the developing cortex.** (A) Principal component analysis of biological replicates from SLAM-seq half-life data. (B) Correlation between z-score normalized half-lives at E11.5 and E14.5. Red dashed line indicates y = x line; r represents the Pearson correlation coefficient. (C) As in B, for E14.5 and E16.5. (D and E) GO analysis of top 10% most stable and least stable transcripts. Circle size represents fold enrichment and color represents adjusted *p*-value. (F) Comparison of 3′ UTR lengths of the top 10% most and least stable transcripts at each stage. (G) As in E, comparing percentage GC content. (H, I) As in F, G, for 5′ UTRs. (J) Comparison of CSC values between E11.5 and E14.5; r represents the Pearson correlation coefficient. (K) As in J, for E14.5 and E16.5. ***$p < 0.001$. Wilcoxon rank-sum test (F, H), Welch's two-sample *t* test (G, I). Underlying data for this figure can be found in S9 Data.
(TIF)

**S2 Fig. Developmentally regulated genes have cell type enriched expression patterns.** (A) Expression of differentially expressed genes at E14.5 compared to E11.5. Expression data from publicly available scRNA-seq data sets [6]. (B) As in A, for differentially expressed genes at E16.5 compared to E14.5. (C) Expression of progenitor-enriched genes across development using SLAM-seq expression data from this study. (D) As in C, for neuron-enriched genes. *$p < 0.05$, ***$p < 0.001$. Wilcoxon rank-sum test (A, B), one-way ANOVA with Tukey's HSD post hoc test (C, D). Underlying data for this figure can be found in S10 Data.
(TIF)

**S3 Fig. *Cnot3* is required for neuronal lamination.** (A) Expression of *Cnot1*, *Cnot2*, and *Cnot3* in E12.5 cortical lysates measured by RT-qPCR. (B–D) Distribution of indicated marker(s) at E18.5 across 5 equally sized cortical bins spanning the ventricular surface (Bin 1) to the pia (Bin 5). *$p < 0.05$, **$p < 0.01$, ***$p < 0.001$. One-way ANOVA with Tukey's HSD post hoc test (A), two-way ANOVA with Tukey's HSD post hoc test (B–D). Error bars represent standard deviation. Underlying data for this figure can be found in S11 Data.
(TIF)

**S4 Fig. *Cnot3* alters RGC distribution and is required for survival of post mitotic neurons.** (A) Immunofluorescence against SOX2 and PAX6 in E14.5 control and *Emx1*-Cre cKO cortices. (B) Schematic showing strategy for *Cnot3* cKO in neurons using *Nex*-Cre. (C) Immunofluorescence against CC3 in E18.5 control and *Nex*-Cre cKO cortices. (D) Immunofluorescence against indicated marker(s) in E18.5 control and *Nex*-Cre cKO cortices. (E–H) Quantification of density for the indicated markers (*n* = 3 embryos per genotype). *$p < 0.05$,**$p < 0.01$. One-way ANOVA with Tukey's HSD post hoc test (F–H). Error bars represent standard deviation.

Scale bars: 50 μm for all images. Underlying data for this figure can be found in S13 Data.
(TIF)

**S5 Fig. *Cnot3;Trp53* dcKO brains have no apoptosis but exhibit some altered lamination patterns.** (A) Representative images of immunofluorescence against CC3 in E12.5 cortices showing rescue of apoptosis in dcKO mice. (B) Representative images of immunofluorescence against p53 in E12.5 cortices showing rescue of p53 accumulation in dcKO mice. (C–F) Distribution of indicated marker(s) at E18.5 across 5 equally sized cortical bins spanning the ventricular surface (Bin 1) to the pia (Bin 5). (G) Representative images showing E12.5 cortices of the indicated genotype pulse labeled by IP injection of BrdU 2 h prior to dissection. (H) Quantificaiton of BrdU+ cells ($n$ = 3–4 embryos per genotype). $^*p < 0.05$, $^{**}p < 0.01$, $^{***}p < 0.001$. Three-way ANOVA with Tukey's HSD post hoc test (C–F). One-way ANOVA with Tukey's HSD post hoc test (H). Error bars represent standard deviation. Scale bars: 50 μm (A, B, G). Underlying data for this figure can be found in S13 Data.
(TIF)

**S6 Fig. *Cnot3*-dependent regulation of gene expression.** (A) Expression of the indicated cell cycle genes at E12.5 measured by RNA-seq. (B) Expression levels of transcripts that are down-regulated in both *Cnot3* cKO and dcKO in either progenitors or neurons. Expression data taken from ref. [6]. (C) As in (B), for transcripts that are up-regulated in both cKO and dcKO cortices. (D) Transcript levels at E12.5 measured by RNA-seq. (E) Decay curves measured by RT-qPCR following transcriptional shut-off in E12.5 primary cultures. Dashed lines indicate fit to exponential decay equation. $^*p < 0.05$, $^{**}p < 0.01$, $^{***}p < 0.001$. Values shown are adjusted $p$-values from DESeq2 (A, D). Error bars represent standard deviation. Underlying data for this figure can be found in S14 Data.
(TIF)

**S1 Table. SLAM-seq half-life and expression data for E11.5, E14.5, and E16.5.**
(XLSB)

**S2 Table. List of transcripts with variable half-lives: related to Fig 2.**
(XLSX)

**S3 Table. Differential expression analysis between developmental stages.**
(XLSB)

**S4 Table. List of genes enriched in progenitors or neurons.**
(XLSX)

**S5 Table. Differentially expressed genes in E12.5 cKO and dcKO cortices.**
(XLSB)

**S6 Table. Primers used in this study.**
(XLSX)

**S7 Table. List of statistical analyses.**
(XLSX)

**S1 Movie. Representative proliferative division: related to Fig 7.**
(AVI)

**S2 Movie. Representative neurogenic division: related to Fig 7.**
(AVI)

**S1 Data. Numerical values underlying graphs in Fig 1.**
(XLSX)

**S2 Data. Numerical values underlying graphs in Fig 2.**
(XLSX)

**S3 Data. Numerical values underlying graphs in Fig 3.**
(XLSX)

**S4 Data. Numerical values underlying graphs in Fig 4.**
(XLSX)

**S5 Data. Numerical values underlying graphs in Fig 5.**
(XLSX)

**S6 Data. Numerical values underlying graphs in Fig 6.**
(XLSX)

**S7 Data. Numerical values underlying graphs in Fig 7.**
(XLSX)

**S8 Data. Numerical values underlying graphs in Fig 8.**
(XLSX)

**S9 Data. Numerical values underlying graphs in S1 Fig.**
(XLSX)

**S10 Data. Numerical values underlying graphs in S2 Fig.**
(XLSX)

**S11 Data. Numerical values underlying graphs in S3 Fig.**
(XLSX)

**S12 Data. Numerical values underlying graphs in S4 Fig.**
(XLSX)

**S13 Data. Numerical values underlying graphs in S5 Fig.**
(XLSX)

**S14 Data. Numerical values underlying graphs in S6 Fig.**
(XLSX)

**S1 Raw Images. Uncropped version of western blot images in Fig 4C.**
(PDF)

## Acknowledgments

We thank members of the Silver and Hu labs for helpful discussions and careful reading of the manuscript.

## Author Contributions

**Conceptualization:** Lucas D. Serdar, Guang Hu, Debra L. Silver.

**Data curation:** Lucas D. Serdar, Jacob R. Egol, Brad Lackford.

**Formal analysis:** Lucas D. Serdar, Jacob R. Egol, Brian D. Bennett, Guang Hu.

**Funding acquisition:** Lucas D. Serdar, Guang Hu, Debra L. Silver.

**Investigation:** Lucas D. Serdar, Brad Lackford, Guang Hu.

**Methodology:** Lucas D. Serdar, Brad Lackford, Debra L. Silver.

**Project administration:** Debra L. Silver.

**Supervision:** Guang Hu, Debra L. Silver.

**Visualization:** Lucas D. Serdar.

**Writing – original draft:** Lucas D. Serdar, Debra L. Silver.

**Writing – review & editing:** Lucas D. Serdar, Jacob R. Egol, Brad Lackford, Brian D. Bennett, Guang Hu, Debra L. Silver.

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
