## [Editor Report · Decision Letter 0]

19 Aug 2024

Dear Dr Silver, 

Thank you for submitting your manuscript entitled "mRNA stability fine tunes gene expression in the developing cortex to control neurogenesis" for consideration as a Research Article by PLOS Biology. Please accept my apologies for the delay in getting back to you last week as we consulted with an academic editor about your submission. 

Your manuscript has now been evaluated by the PLOS Biology editorial staff, as well as by an academic editor with relevant expertise, and I am writing to let you know that we would like to send your submission out for external peer review.

Once your full submission is complete, your paper will undergo a series of checks in preparation for peer review. After your manuscript has passed the checks it will be sent out for review. To provide the metadata for your submission, please Login to Editorial Manager (https://www.editorialmanager.com/pbiology) within two working days, i.e. by Aug 21 2024 11:59PM.

Kind regards,

Richard

Richard Hodge, PhD

rhodge@plos.org

PLOS

---

## [Decision Letter · Decision Letter 1]

14 Oct 2024

Dear Dr Silver,

Thank you for your patience while your manuscript "mRNA stability fine tunes gene expression in the developing cortex to control neurogenesis" was peer-reviewed at PLOS Biology. Please accept my sincere apologies for the delays that you have experienced during the peer review process. Your manuscript has now been evaluated by the PLOS Biology editors, an Academic Editor with relevant expertise, and by two independent reviewers. 

In light of the reviews, which you will find at the end of this email, we would like to invite you to revise the work to thoroughly address the reviewers' reports.

As you will see below, the reviewers are generally positive about your study and note that the work is well done and important. Reviewer #1 raises some concerns about the inclusion of the dataset presented in Figure 8 and asks that a discussion of the functional relevance of the identified transcripts is provided. In addition, Reviewer #2 asks that a control analysis of cortical thickness in the CNOT3-heterozygous mice is included, as well as providing quantifications for CC3-positive cells in the control cortex.

Given the extent of revision needed, we cannot make a decision about publication until we have seen the revised manuscript and your response to the reviewers' comments. Your revised manuscript is likely to be sent for further evaluation by all or a subset of the reviewers.

**IMPORTANT - SUBMITTING YOUR REVISION**

*Re-submission Checklist*

*Published Peer Review*

*PLOS Data Policy*

*Blot and Gel Data Policy*

Sincerely,

Richard

Richard Hodge, PhD

rhodge@plos.org

REVIEWS:

Reviewer #1: The Silver lab has a long and impressive history in studying cortical development with a special emphasis on neural progenitors. In the current paper, they take on to systematically identify the landscape of RNA stability in the developing task using SLAM-seq, a by now well established approach in RNA biology. This allowed them to identify the most and least stable transcripts, respectively, for each time point of mouse development and subsequently analyze their 3'-UTR in detail. Rewardingly, but not surprisingly, this careful and well-planned analysis yielded distinct clusters of transcripts with different dynamics at different time points, which is the promising basis for this important study. The importance and the quality of this data set is clearly shown in the provided Volcano plots in Fig. 3, identifying known markers for neuronal development, e.g. Neurod2, Dcx, Hes5, Sox2. Also, progenitor-enriched transcripts are often rather unstable compared to their neuron-enriched counterparts arguing for cell-specific layers of RNA stability. Next, the authors turned to CNOT3, a central component of the CCR4-NOT complex important for deadenylation in cells, with a known role in neurodevelopmental diseases. Taking advantage of their impressive mouse genetics expertise, they decided to target radial glia and their progeny, e.g. IPCs and neurons, by using the existing Emx1-Dre driver line and characterized the resulting transgene in great detail. Interestingly, their cKO mice exhibited profound microcephaly with significant reduction in cortical thickness, with a particular impact on glutamatergic neurons. In particular, Cnot3 cKO mice have early defects during mid-neurogenesis arguing for an early role of CNOT3 affecting both RGCs and IPCs. Consequently, apoptosis was found to take place in these mice as judged by cleaved caspase 3 staining and by accumulation of p53, an important signaling molecule for apoptosis. Rescue of p53-dependent apoptosis led to a clear rescue of cortical size. Consistent with this is their finding that the loss of upper layer neurons (but not their lower layer counterpart) is dependent on p53. Using a clever in vivo semicumulative labeling assay, they found that Cnot3 loss yielded an overall slower cell cycle consistent with their previous findings. Most impressively, when performing a live imaging assay to monitor cell fates showed that progenitors undergo fewer neurogenic divisions in CKO cells arguing for impaired neuron generation. Obviously, the authors wanted to know on which targets CNOT3 might act and therefore performed bulk RNA-seq from their various mouse lines yielding large lists of potential RNA targets. Here, there is evidence that CNOT3 is likely to control expression of poorly expressed, non-optimal mRNAs (based on codon usage) in the cortex. 

Overall, the work is of usual high quality, very important for a broad community with important biological implications. The data in the presented figures are impressive, important and the experiments well described. Please find some comments for the authors to consider before publication. 

I do not know about the history of this important study. It may have seen the eyes of reviewers before and possibly their suggestions have impacted on the current manuscript. I am not sure that the inclusion of Figure 8 really helps the consistency of the study. Needless to say that the identification of important targets of CNOT3 would be a big advance. However, I am not convinced that the current list of data is consistent, logical and of functional importance yet. If this set of data stays in, the authors might want to focus on telling us, which of the identified transcripts might be functionally relevant here. At least, a more coherent view on what has been done would help the reader. 

Further comments for the authors

1. Enlarge panel A in Fig. 1 to do justice to this important scheme

2. Possibly combine Figs 2. And 3, e.g. the key panels. 

3. Provide your rationale for why to focus on CNOT3! Indicate this gene in your previous plots and tell us the evidence? 

4. Improve content of scheme Fig. 4A

5. Show video (or a sequence of stills) from live cell imaging data in Fig. 8

Reviewer #2 (Yi-Shuian Huang, signs review): Serdar et al. reported transcriptome-wide alterations in RNA stability during cortical development and highlighted how molecular defects in RNA degradation contribute to abnormal cortical development in CNOT3-cKO mice. Through metabolic labeling of cells isolated from cortices at E11.5, E14.5, and E16.5, their SLAM-seq analysis uncovered several key omics principles, including that increased RNA stability is correlated with shorter 5'-UTR and 3'-UTR lengths, reduced m6A modifications, and higher usage of optimal codons. Although dysregulation of RNA turnover is associated with neurodevelopmental disorders, its specific role in cortical development remains largely unclear. Given that mutations in CNOT1 and CNOT3, both components of the CCR4-NOT complex, are associated with autism and developmental disabilities, the authors opted for conditional ablation of CNOT3 in Emx1-lineage cells to investigate the resulting defects in cortical development. Notably, RNAs that are upregulated during development tend to exhibit greater stability, while those that are downregulated are generally less stable. Together, their findings demonstrate that precise regulation of RNA turnover is essential for proper brain development. This is a well-done and excellent study. I only have a few questions I would like the authors to clarify. 

Specific Comments:

1. What is the cortical size in CNOT3-heterozygous mice? To better reflect the patient situation, one would expect cortical development to be impaired in these mice. Can the authors provide at least a basic analysis of cortical thickness and layer thickness in CNOT3-heterozygous mice? If this analysis is not feasible within the revision timeline (assuming they do not keep global heterozygous knockout mice), further discussion connecting their findings to clinical patients should be elaborated.

2. In Fig 5E, there are only a few CC3-positive cells, but they should be detected in the control cortex. Could the authors provide quantified data for this, since they have images already?

3. Interestingly, the conditional ablation of p53 rescues neurons in the superficial layers but not in layers 5-6. The authors suggest that the loss of upper layer neurons is p53-dependent, whereas the loss of deep layer neurons is largely p53-independent. Could it be possible that proliferation plays a more significant role than apoptosis in the deeper layer neurons? If so, what is the number of BrdU-labeled cells in E12.5 dcKO cortex compared to those in Ctrl and CNOT3-cKO cortices shown in Fig. 7A?

4. Please deposit the NGS data and provide the GSE number in the revised manuscript.

---

## [Editor Report · Decision Letter 2]

9 Jan 2025

Dear Dr Silver,

Thank you for your patience while we considered your revised manuscript "mRNA stability fine-tunes gene expression in the developing cortex to control neurogenesis" for publication as a Research Article at PLOS Biology. Please accept my apologies for the delay in getting back to you with feedback due to the recent Christmas holidays. This revised version of your manuscript has been evaluated by the PLOS Biology editors and the Academic Editor.

Based on our Academic Editor's assessment of your revision, I am pleased to say that we are likely to accept this manuscript for publication, provided you satisfactorily address the following data and other policy-related requests that I have provided below (A-E):

(A) In the animal ethics statement in the Methods section, please provide the specific approval number issued by Duke Institutional Animal Care and Use Committee to conduct the mouse studies.

(B) Thank you for depositing the sequencing data in the GEO database (GSE281690 and GSE281693). However, we note that the data is currently on hold for release. We ask that you please make the sequencing data publicly available at this stage before publication.

(C) Please also ensure that each of the relevant figure legends in your manuscript include information on *WHERE THE UNDERLYING DATA CAN BE FOUND*, and ensure your supplemental data file/s has a legend.

(D) Per journal policy, if you have generated any custom code during the course of this investigation, please make it available without restrictions. Please ensure that the code is sufficiently well documented and reusable, and that your Data Statement in the Editorial Manager submission system accurately describes where your code can be found. 

(E) Please ensure that your Data Statement in the submission system accurately describes where your data can be found and is in final format, as it will be published as written there. 

We expect to receive your revised manuscript within two weeks. 

*Published Peer Review History*

*Press*

Best wishes,

Richard

Richard Hodge, PhD

rhodge@plos.org

PLOS

---

## [Editor Report · Decision Letter 3]

23 Jan 2025

Dear Debra,

On behalf of my colleagues and the Academic Editor, Bassem Hassan, I am pleased to say that we can accept your manuscript for publication, provided you address any remaining formatting and reporting issues. These will be detailed in an email you should receive within 2-3 business days from our colleagues in the journal operations team; no action is required from you until then. Please note that we will not be able to formally accept your manuscript and schedule it for publication until you have completed any requested changes.

PRESS

Best wishes, 

Richard

Richard Hodge, PhD

rhodge@plos.org

PLOS
